# Oncogenic mutations of PIK3CA lead to increased membrane recruitment driven by reorientation of the ABD, p85 and C-terminus

Meredith L. Jenkins[1], Harish Ranga-Prasad[1], Matthew A. H. Parson [1], Noah J. Harris[1], Manoj K. Rathinaswamy[1] & John E. Burke [1,2] ✉

*PIK3CA* encoding the phosphoinositide 3-kinase (PI3K) p110α catalytic subunit is frequently mutated in cancer, with mutations occurring widely throughout the primary sequence. The full set of mechanisms underlying how PI3Ks are activated by all oncogenic mutations on membranes are unclear. Using a synergy of biochemical assays and hydrogen deuterium exchange mass spectrometry (HDX-MS), we reveal unique regulatory mechanisms underlying PI3K activation. Engagement of p110α on membranes leads to disengagement of the ABD of p110α from the catalytic core, and the C2 domain from the iSH2 domain of the p85 regulatory subunit. PI3K activation also requires reorientation of the p110α C-terminus, with mutations that alter the inhibited conformation of the C-terminus increasing membrane binding. Mutations at the C-terminus (M1043I/L, H1047R, G1049R, and N1068KLKR) activate p110α through distinct mechanisms, with this having important implications for mutant selective inhibitor development. This work reveals unique mechanisms underlying how PI3K is activated by oncogenic mutations, and explains how double mutants can synergistically increase PI3K activity.

Activating mutations in the gene *PIK3CA*, which encodes the class IA phosphoinositide 3-kinase (PI3K) catalytic subunit p110α are among the most common mutations across human cancers[1–4]. Class I PI3Ks are composed of four isoforms (class IA [p110α, p110β, p110δ] and class IB [p110γ]), which together generate the lipid second messenger phosphatidylinositol 3,4,5 trisphosphate (PIP$_3$) downstream of multiple cell surface receptors, including receptor tyrosine kinases (RTKs), G-protein coupled receptors (GPCRs) and Ras superfamily GTPases. PIP$_3$ recruits multiple effectors, including kinases and regulators of G protein signalling which play critical roles in regulating growth, survival, proliferation and metabolism[5–7].

PI3Kα is a heterodimer composed of a p110α catalytic subunit and a p85 regulatory subunit, of which there are 5 isoforms: p85α, p55α and p50α (encoded by *PIK3R1*), p85β (encoded by *PIK3R2*) and p55γ (encoded by *PIK3R3*). The class IA regulatory subunits play three key roles in regulating class IA PI3Ks: they stabilise and inhibit the p110

catalytic subunits[8,9], and they allow for activation through the direct engagement of the nSH2 and cSH2 domains of regulatory subunits with pYXXM motifs in RTKs[10]. In unstimulated cells, the p110α protein is kept in an inactive and stable cytosolic configuration due to its interactions with the regulatory subunit, with full activation and membrane recruitment of the PI3Kα complex requiring binding to both GTP loaded Ras, and engagement of both the nSH2 and cSH2 of regulatory subunits by bis-phosphorylated pYXXM motifs present in RTKs and their adaptors[11–13]. However, the full molecular mechanisms underpinning activation and membrane binding of class IA PI3K, and how oncogenic mutants alter this are not completely understood.

Extensive biochemical and biophysical studies have revealed how p110α is inhibited by regulatory subunits[14–20]. The p110α catalytic subunit is composed of five domains (an Adaptor Binding Domain (ABD), Ras Binding Domain (RBD), C2, helical and a bi-lobal kinase domain)[14,21] (Fig. 1A). The RBD, C2, helical and kinase domains

[1]Department of Biochemistry and Microbiology, University of Victoria, Victoria, BC V8W 2Y2, Canada. [2]Department of Biochemistry and Molecular Biology, The University of British Columbia, Vancouver, BC V6T 1Z3, Canada. ✉e-mail: jeburke@uvic.ca

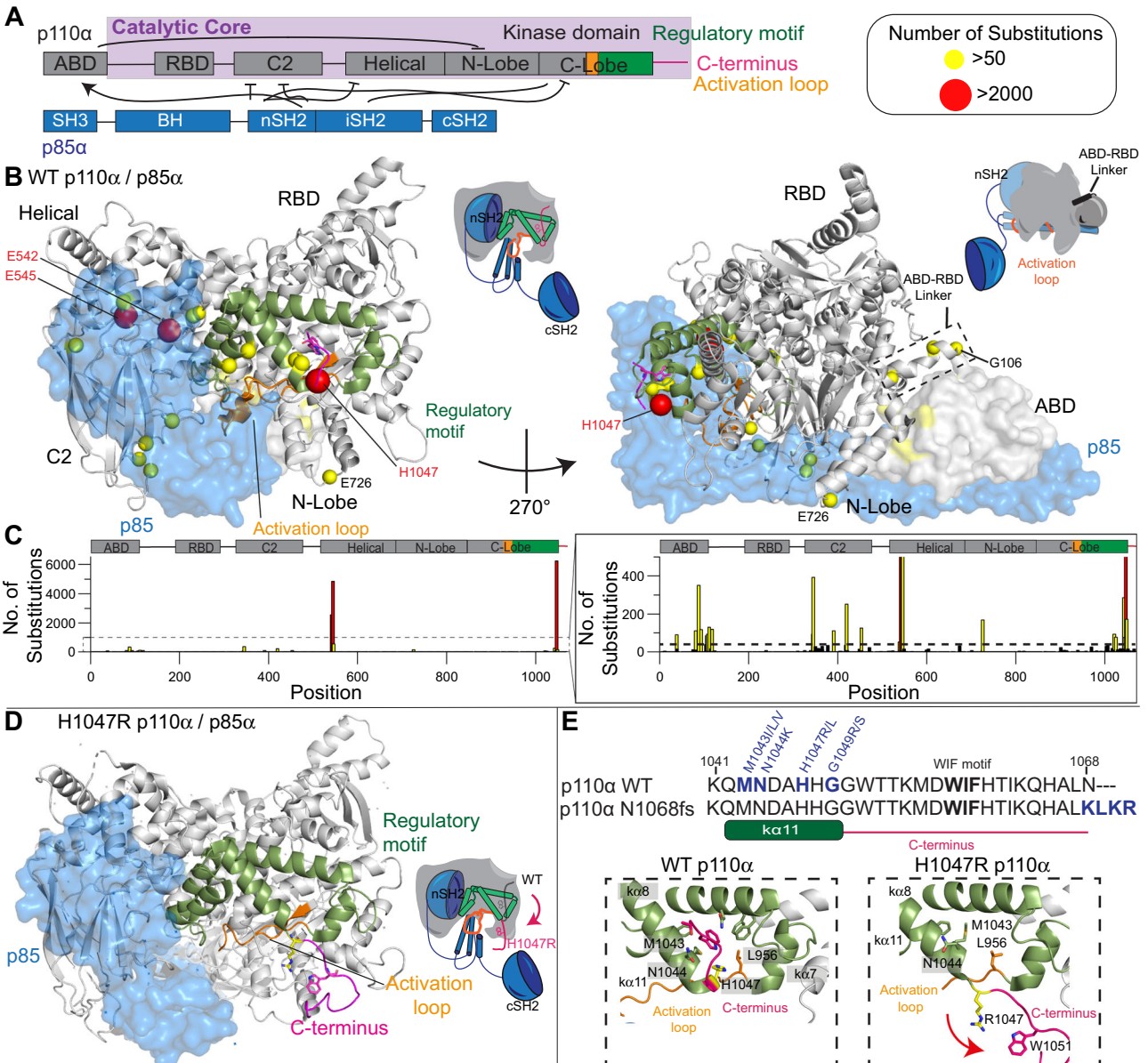

**Fig. 1 | Structure of p110α/p85α and location of *PIK3CA* oncogenic mutations.**
**A** Domain schematic of both p110α/p85α, with the catalytic core, activation loop (orange), regulatory motif (green) and C-terminus (magenta) of p110α annotated. Same colour scheme has been used to map these features on the structures below. **B** Oncogenic mutations in p110α mapped on the structure of WT p110α/p85α (PDB: 4OVU[39]). The frequency of oncogenic mutations from the COSMIC database[44] as described in panel **C** is coloured according to the legend with any mutation with a frequency >50 indicated as a sphere. Other features are coloured according to the domain schematic in panel **A**. The p85 subunit is shown as a transparent blue surface. Cartoons of the two views of PI3K highlighting these features are shown to the right of each structural model. These same cartoon views are used to map all

further HDX-MS data. **C** Frequency of mutations across the primary sequence of *PIK3CA* from the Catalog of Somatic Mutations in Cancer (COSMIC) database (data from January 2022)[44]. **D** C-terminus in H1047R adopts a unique confirmation compared to WT (PDB: 3HHM)[17]. Features are coloured the same as in panel **B**. **E** Above: the sequence of the C-terminus is shown, with mutants coloured blue, and the membrane binding WIF motif in bold[57]. The sequence of a frameshift mutant (N1068KLKR)[30] is also shown. Below: Orientation of the C-terminus in the WT (PDB: 4OVU) and H1047R (PDB: 3HHM) crystal structures. The relative positioning of additional oncogenic mutants (M1043I/L, N1044K, G1049R/S) are indicated. The reorientation of the C-terminus (coloured in magenta) that occurs upon H1047R mutation is indicated by the red arrow.

together form the catalytic core of PI3Ks, which is conserved through the class I, II and III PI3Ks[22,23]. The ABD of class IA PI3Ks binds irreversibly to the iSH2 coiled coil domain present in all class IA regulatory subunits[15], and also forms an inhibitory intra-subunit interface with the N-lobe of the kinase domain (Fig. 1A, B). The iSH2 also makes inhibitory contacts with the C2 and activation loop of the kinase domain[16,24]. The nSH2 domain of regulatory subunits binds reversibly to the C2, helical and kinase domains[15,17], with these contacts broken when the nSH2 binds to pYXXM motifs[19]. The cSH2 of regulatory subunits is strictly required for high affinity association

with bis-phosphorylated receptors and their adaptors[11], and in p110β and p110δ forms an inhibitory interface with the kinase domain, however, this interaction is either absent or transient in p110α[25,26]. Recently Cryo-EM analysis of p110α bound to p85α showed that upon binding to phosphopeptides there appeared to be complete disengagement of the ABD and regulatory subunit from the catalytic core[27], which was not fully supported by previous HDX-MS analysis of phosphopeptide binding[16,19]. However, disengagement of the catalytic core from the ABD and regulatory subunits is consistent with HDX-MS results studying PI3K membrane binding[12].

The *PIK3CA* and *PIK3R1* genes encoding p110α and p85α are both frequently mutated in human cancers[1,2,28]. The majority of mutations lead to amino acid substitutions (Fig. 1C)[29], although, more complex insertions and deletions also occur[30]. Oncogenic transformation by p85α mutations is driven by activation of the p110α isoform[31]. For p110α the most frequent mutations are located at two hot spots located at the helical-nSH2 interface (E542K, E545K) and the C-terminus of the kinase domain (H1047R) (Fig. 1B). The E545K disrupts the nSH2-helical interface[15] while H1047R alters the membrane binding surface of the kinase domain, and how the kinase domain packs against the activation loop (Fig. 1D, E)[17,32]. However, there also are relatively high-frequency mutations at the ABD-kinase interface, the ABD-RBD linker, the C2-iSH2/C2-nSH2 interfaces, the putative membrane interface of the N-lobe with membranes, and in a region of the kinase domain C-terminal to the activation loop, referred to as the regulatory motif (Fig. 1C)[33]. We have previously found that oncogenic mutations mimicked and enhanced conformational changes observed in the catalytic cycle of WT PI3Kα[16], with different mutants showing unique conformational changes. Different *PIK3CA* mutations activating through unique mechanisms is supported by the discovery of tumours harbouring double *PIK3CA* mutations *in cis*, with these tumours showing enhanced sensitivity to PI3K inhibition[34].

Here, we show using a combination of hydrogen deuterium exchange mass spectrometry (HDX-MS) and kinase/membrane binding assays a molecular model for how oncogenic mutants in p110α activate both kinase activity and membrane binding. This reveals the critical role of disengagement of the inhibitory contacts of the ABD domain and the p85 regulatory subunit, as well as the importance of reorganisation of the membrane binding C-terminus in PI3K activation. Intriguingly, mutations at the C-terminal tail activate PI3K through distinct molecular mechanisms which provides insight into how this might be utilised for design of mutant selective inhibitors. Overall, this work provides unique insight into the molecular mechanisms mediating PI3K activation by oncogenic mutations.

## Results

To investigate the role of the ABD domain/p85 regulatory subunit in controlling PI3K enzyme activity, we needed a construct that allowed us to interrogate the dynamic effects of full ABD disengagement. We engineered and purified the catalytic core of p110α (106–1068, referred to as the p110α core) along with the full-length complex of p110α-p85α (full set of all constructs purified in this manuscript shown in Supplementary Fig. 1, and SDS-page gels in Supplementary Fig. 2). Initial attempts to purify the p110α core construct were unsuccessful, and we were only able to successfully purify this construct when it contained a kinase dead mutation (D915N). To validate that the D915N p110α construct did not cause any significant changes in protein conformation or membrane binding, we carried out HDX-MS experiments on WT and kinase dead p110α-p85α complexes bound to a PDGFR bis-phosphorylated pY peptide (referred to going forward as pY) composed of PDGFR residues 735–767 with phosphorylation present at Y740 and Y751 in the presence and absence of membrane vesicles (5% PIP$_2$, 30% PS, 65% PE, referred to afterwards as PIP$_2$/PS/PE). HDX-MS is a technique that measures the exchange rate of amide hydrogens, and as the rate is dependent on the presence and stability of secondary structure, it is an excellent probe of protein conformational dynamics, and we have extensively used it to study PI3K activation[11,12,16,19,25]. There were no significant changes in conformation in the kinase dead p110α compared to wild type, and both had equivalent membrane binding to lipid membranes (Supplementary Fig. 3), highlighting the suitability of this mutant for membrane binding and HDX-MS studies.

We carried out HDX-MS experiments to identify conformational changes that occur in the catalytic core of p110α upon

removal of the ABD and p85α regulatory subunit. We compared H/D exchange differences for both the p110α core and the full-length complex of p110α-p85α. The full details of HDX-MS data processing are in Table S1, with all raw HDX-MS data for all time points available in the source data. For the p110α core construct there was significantly increased exchange (for all HDX-MS experiments this is defined as differences at any time point >5%, >0.4 Da, and a *p*-value <0.01 for a two-tailed *t*-test) in the ABD-RBD linker (114–119), C2 domain (347–355 and 444–475), helical domain (524–551), N-lobe of the kinase domain (691–697, 735–744), the activation loop (930–937) and the C-lobe of the kinase domain (1002–1013) (Fig. 2A–D and S4A, B). All significant differences in the catalytic core of p110α were in regions that were either in contact with ABD or p85α. This dataset allowed us to compare changes to those we previously observed upon membrane recruitment by both pY, and membrane bound Ras of the full-length p110α-p85α complex (Fig. 2B)[12]. Intriguingly, we find that the region of the ABD-RBD linker in contact with the ABD domain (114–119) had similar increases in exchange between WT p110α-p85α and either the p110α core or upon pY/HRas mediated membrane recruitment (Fig. 2D). There is a small increase in exchange in this region upon pY binding alone, but this was minor compared to the effect upon membrane binding[16,19]. The increase in region 114–119 was greater when bound to pY/HRas membranes compared to pY-mediated membrane binding alone, suggesting this increase is dependent on the amount of membrane binding (Fig. 2D). This is also the case for increases in exchange observed upon membrane binding in the N-terminal and C-terminal ends of the iSH2 domain that are in contact with the C2 domain (470–476, 556–570, Supplementary Fig. 4E), showing the clear link between increases in the ABD-RBD linker and at the C2-iSH2 interface.

This data comparing the full-length heterodimer vs p110α core allowed us to define the effect of ABD removal on the contact site at the ABD-RBD linker. This region still is protected from exchange at early time points, suggesting presence of secondary structure, however, it is much more dynamic in the absence of the ABD. Comparing this to previous HDX-MS experiments examining pY-Ras membrane recruitment of p110α -p85α[12], showed that the exchange rate of the core is similar to the p110α-p85α membrane bound state, suggesting a correlative ABD disengagement occurring with membrane binding. This is supported by our previous observation of increased membrane binding for oncogenic mutants at the C2-iSH2 or ABD interfaces (N345K, G106V and G118D) that would be expected to promote ABD / iSH2 disengagement[16]. An important note is that this data does not support complete dissociation of the p110-p85 complex (due to the extremely high affinity interaction of the ABD to the iSH2[35]), but instead the ABD-p85 becoming mobile relative to the p110α catalytic core.

### Enhanced membrane binding of p110α catalytic core

Our hypothesis that disengagement of the ABD and the regulatory subunit p85α subunit is required for membrane binding suggested that there should be differential membrane binding of the p110α core compared to full-length p110α/p85α. We used protein-lipid Fluorescent Resonance Energy Transfer (FRET) assays (Fig. 3A) to compare membrane recruitment of p110α core to full-length p110α-p85α in the presence and absence of 1 μM pY. This assay was carried out on two different lipids: one optimised for maximal PI3K recruitment (5% PIP$_2$, 10% Dansyl-PS, 25% PS and 60% PE, Fig. 3B), and another roughly mimicking the plasma membrane (5% PIP$_2$, 10% Dansyl PS, 15% PS, 40% PE, 15% PC, 10% cholesterol and 5% sphingomyelin, Supplementary Fig. 4E). While pY was required for robust binding of full-length p110α-p85α, it was dispensable for p110α core association, which is expected

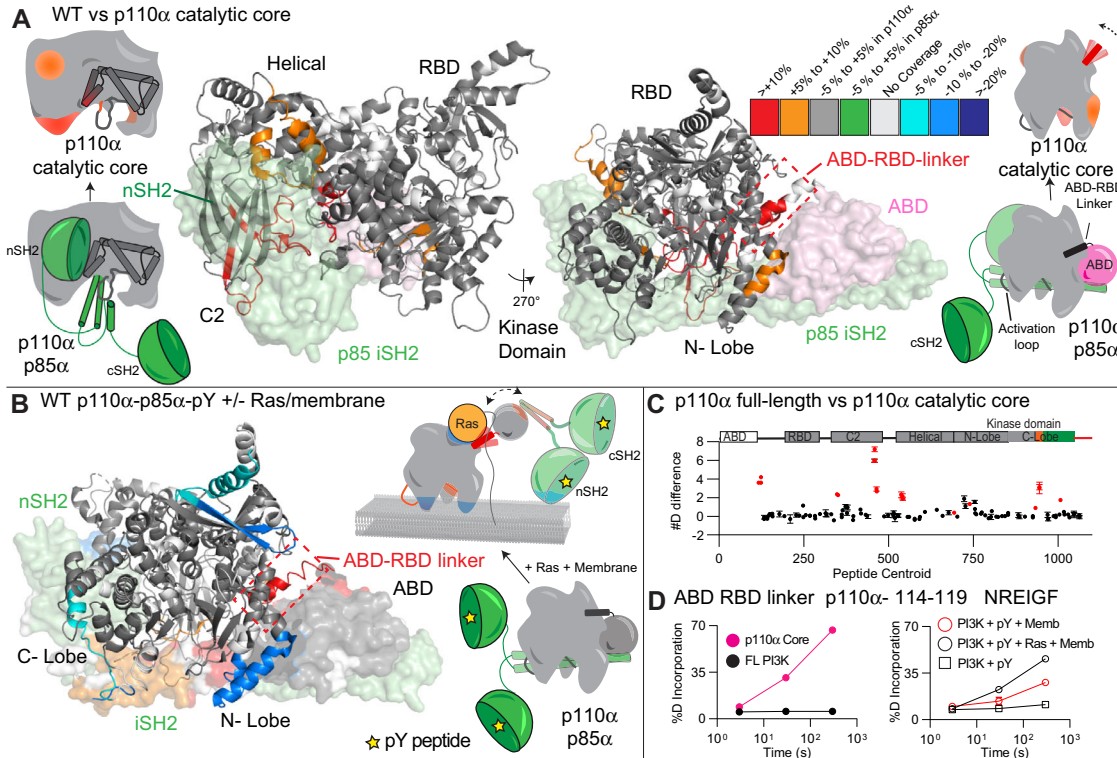

**Fig. 2 | Conformational changes in p110α core compared to full-length p110α-p85α, and comparison to changes upon pY/Ras membrane recruitment.**
**A** Peptides in p110α that showed significant differences in HDX (>0.4 Da and 5% difference, with a two-tailed *t*-test *p* < 0.01) between the catalytic core and the full-length complex are mapped on the structure of p110α-p85α complex (PDB: 4OVU [https://www.rcsb.org/structure/4ovu].) according to the legend. The regions of the ABD (pink) and p85 (green) that are missing in the p110α core are shown as a surface. Cartoon models representing the differences between states are shown next to the structures. A more extensive set of peptides are shown in Supplementary Fig. 4. **B** Peptides in p110α that showed significant differences in both p110α and p85α between free (with pY) and membrane-bound (pY, membrane Ras) (data adapted from Siempelkamp et al.[12]) are mapped onto the structure of p110α-p85α (PDB: 4OVU) according to the legend. Data are presented as the mean, with

error bars representing SD (*n* = 3). **C** The sum of the number of deuteron difference for all peptides analysed over the entire deuterium exchange time course for p110α core compared to full-length p110α- p85α. Peptides coloured in red are those that had a significant change (>0.4 Da and 5% difference at any timepoint, with a two-tailed *t*-test *p* < 0.01). Each point represents a single peptide, and error bars are shown as the sum of S.D. across all time points (*n* = 3 for each time point).
**D** Selected p110α peptide in the ABD-RBD linker that showed increases in exchange in p110α core compared to full-length p110α-p85α (left), and upon membrane binding of full-length PI3K (right, data adapted from Siempelkamp et al.[12]). (Mean is shown, with error bars representing S.D., *n* = 3)., with smaller than the size of the point. A more extensive set of peptides comparing the full-length p110α-p85α with p110α core are shown in Supplementary Fig. 4, with the full list of all peptides and their deuterium incorporation in the source data file.

---

due to the lack of SH2 domains required for pY binding (Fig. 3B). The p110α core showed increased membrane recruitment for both lipid mixtures compared to the pY activated p110α/p85α complex. To determine the role of free p85α in PI3K membrane recruitment, we also purified recombinant free p85α and analysed the protein-lipid FRET signal. There was a weak FRET signal for p85α alone, with only a small change upon pY binding (Fig. 3B and Supplementary Fig. 4F). This signal was significantly lower than the p110α/p85α complex, indicating that the p110α catalytic core is fully capable of membrane binding in the absence of p85α.

### Defining the membrane binding surface of p110α core
We have extensively characterised the membrane binding of the p110α/p85α complex using HDX-MS, however, the disengagement of the ABD and p85 from the catalytic core has likely complicated the analysis of membrane binding regions. We carried out HDX-MS experiments of p110α core in the presence and absence of 5% PIP₂/PS/PE membranes to fully understand the molecular under-pinnings of p110α membrane binding. We observed protection in the ABD-RBD linker (114–119), C2 (343–355), N-lobe kinase domain (713–734,735–744,799–811 and 850–859), Activation loop (930–961) and the C-terminus of the C-lobe kinase domain (1039–1055 and 1056–1068) (Fig. 3A–C). The largest differences occurred in the C-terminus, and N-lobe, with only minor

differences in the C2 domain. However, the region of the C2 domain that interacts with membrane has limited secondary structure (see Supplementary Fig. 4D), which can make tracking transient membrane differences using HDX challenging. Previous HDX-MS experiments testing N345K p110α-p85α binding to membranes showed this same region being protected by membranes[16]. Overall, this supports a model where p110α binds membrane at a surface composed of the C2 domain, the kα1–kα2 helices (720–744) and the 859–872 region of the N-lobe, the activation loop, along with the C-terminal tail. Intriguingly, the kα1–kα2 helices in the N-lobe interact with the ABD, and the 343–355 region of the C2 domain binds the N + C termini of the iSH2 domain, which provides a putative molecular explanation for why disengagement of the ABD and p85 leads to increased membrane association.

When comparing our data to the full set of missense onco-genic mutations in the ABD, ABD-RBD linker, C2, helical and the N-lobe of the kinase domain we find that all mutations found in >30 tumours except one (E726K) are located at either the ABD or p85 interfaces. We had previously observed that mutations in the ABD-RBD linker caused similar conformational changes to those located at the C2-iSH2 interface, with this being explained by both muta-tions leading to disengagement of the ABD and p85 from the cat-alytic core[16].

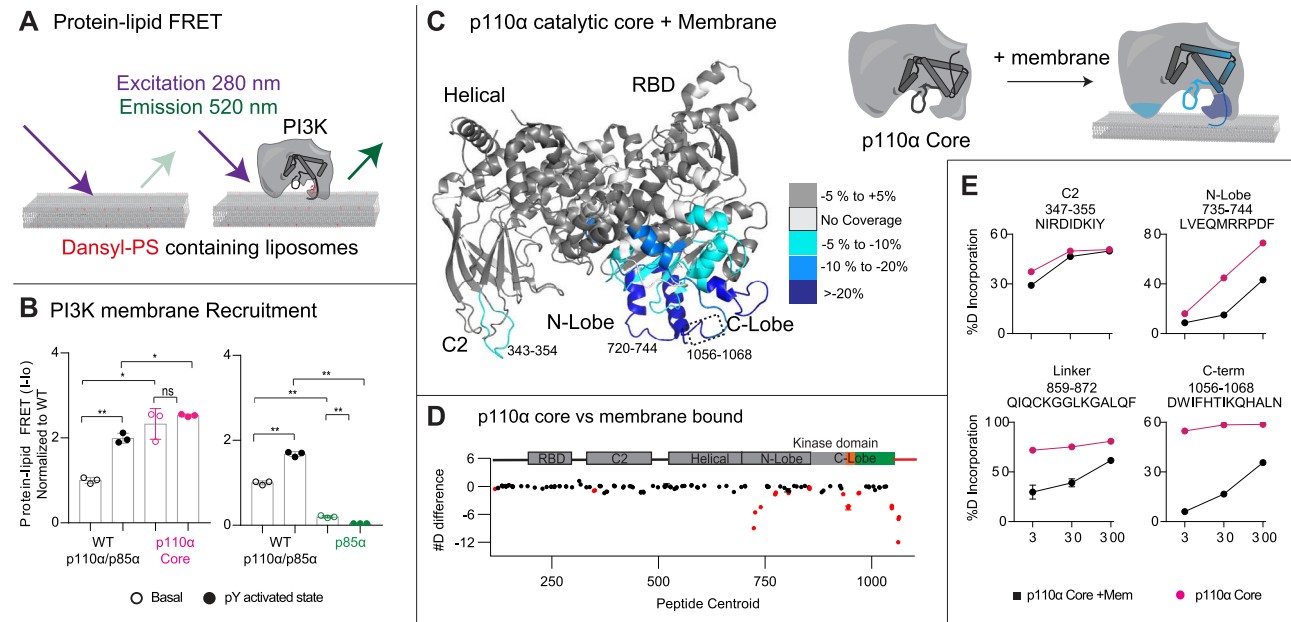

Fig. 3 | Enhanced membrane binding of p110α core compared to full-length p110α/p85α, and mapping of the p110α membrane binding interface. A Cartoon schematic describing the protein-lipid FRET assay, where tryptophan's in the protein are excited at 280 nm, with emission at 350 nm, which upon membrane binding can excite the dansyl moiety in dansyl phosphatidylserine lipids, leading to emission at 520 nm. B Protein-lipid FRET measurements of membrane recruitment comparing p110α core and full-length p110α/p85α complex as well as full-length p110α/p85α complex and p85a apo under basal and pY activated conditions on PE/PS/PIP₂ liposomes containing 5% brain PIP₂, 65% egg yolk PE, 25% brain PS and 10% Dansyl-PS (error bars are S.D., $n = 3$). Experiments were carried out with 1 μM pY, 0.5 μM PI3K and 16.65 μg lipid vesicles. The values were normalised to WT apo. Two-tailed t-test p-values represented by the symbols as follows: **<0.001; *<0.05; n.s.>0.05. C Peptides in p110α core that showed significant differences in HDX (>0.4 Da and 5% difference, with a two-tailed t-test p < 0.01) upon binding to 5% PIP₂/PS/PE vesicles were mapped onto the catalytic core of p110α (PDB: 3HHM) according to the legend. D The sum of the number of deuteron difference for all peptides analysed over the entire deuterium exchange time course for p110α core upon binding membranes. Peptides coloured in red are those that had a significant change (>0.4 Da and 5% difference at any timepoint, with a two-tailed t-test p < 0.01). Each point represents a single peptide, and error bars are shown as the sum of S.D. across all time points ($n = 3$ for each time point). A more complete set of peptides comparing the full-length p110α-p85α with p110α core are shown in Supplementary Fig.4, with the full list of all peptides and their deuterium incorporation in the source data file. E Selected p110α peptides in the kinase domain that showed decreases in exchange in the p110α core upon binding membranes. (Mean is shown, with error bars representing S.D., $n = 3$). Source data for this figure are provided in the Source Data file.

## Biochemical analysis of oncogenic mutations in the inhibitory C-terminus

While the disengagement of the ABD and p85 being involved in membrane binding provides a molecular rationale for activation by oncogenic mutations in the ABD, C2, and helical domains, it does not fully explain the molecular mechanism of mutations in the C-lobe of the kinase domain. Our previous HDX-MS analysis of the kinase domain mutant, H1047R, showed increased exposure throughout regions of the N-lobe and C-lobe of the kinase domain, with this resulting in a dramatic increase in membrane binding[16]. In high-resolution structures the C-terminus of p110α makes extensive contacts with helices that make up the regulatory motif (kα8 and kα11) and sits on top of the C-terminus of the activation loop (Fig. 1E)[36–39]. One of the primary interactions is an extensive interface between Trp1051 of the C-terminus and His1047 as well as hydrophobic residues lining kα8 and kα11 of the kinase domain (Met1043, Phe977,980 and 1039). This region is also in contact with Leu956 in the C-terminal end of the activation loop. This orientation positions the membrane binding WIF motif (residues 1057–1059) distant from the membrane binding interface. When comparing these WT p110α structures to the H1047R crystal structure (3HHM), there is a significant reorganisation of the C-terminus, with all contacts between the C-terminus and kα8 and kα11 disrupted, and the WIF motif oriented at the putative membrane surface (Fig. 1D, E)[17]. However, many features of this structure are likely artefactual, and driven by crystal packing, as recent Cryo-EM structures did not observe this difference[32].

To understand the regulatory mechanisms underlying the inhibitory interface with the C-terminus we analysed the most frequent oncogenic mutants that occur at or near this interface. While H1047R/L is the most frequent mutation (>6000 from the COSMIC database), there are multiple frequent missense mutations in this region, including M1043L/I/V (>300), G1049R/S (>150), and N1044K/S (>80). In addition, we analysed an activating frameshift variant that alters the C-terminus by replacing the terminal N with a KLKR extension, which was recently identified in tumour samples (N1068KLKR, referred to afterwards as N1068fs)[30]. To understand if these mutants were activated in a similar way to H1047R, we purified the four p110α mutant complexes (M1043L, H1047R, G1049R, and N1068fs) all bound to full-length p85α.

We characterised the intrinsic ATPase activity of each p110α mutant (Fig. 4A + B), and while this assay does not measure biologically relevant PIP₃ activity, it can measure intrinsic differences in PI3K activity independent of membrane binding. H1047R, G1049R and M1043L had significantly increased ATPase activity compared to WT (Fig. 4A + B). However, the N1068fs mutants showed no increase in ATPase activity compared to WT. This suggested a possible conformational difference between M1043L, H1047R and G1049R compared to N1068fs.

For these mutants, we had difficulty in obtaining sufficient yield of the proteins for extensive biophysical analysis. To circumvent this, we used the kinase dead variants to characterise their membrane binding using protein-lipid FRET using both PM mimic and optimised binding lipids. Membrane binding was enhanced for mutants upon addition of pY, with greater binding for the optimised lipids over the PM mimic

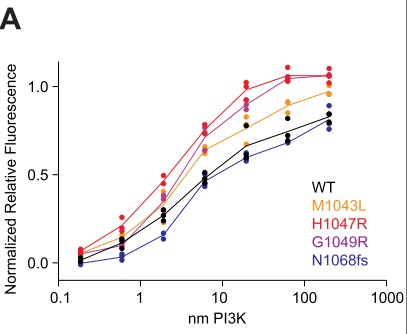
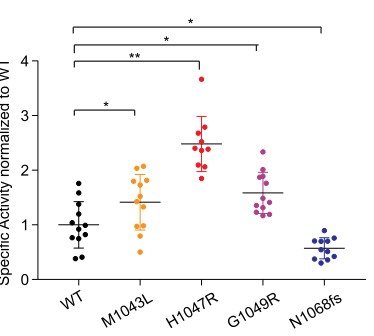
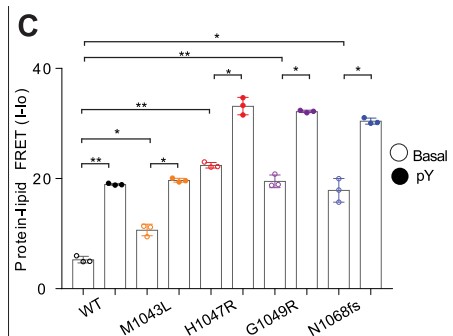

**Fig. 4 | Biochemical analysis of C-terminal *PIK3CA* mutations and their effect on membrane binding and ATPase assays. A** Measurement of ATP turnover performed with different p110α constructs in solution. Experiments were performed with 200 to 0.186 nM PI3K, with 100 μM ATP in the absence of lipid vesicles. **B** Specific activity values normalised to WT for the ATPase assay performed with different p110α constructs in solution (technical replicates, data is presented as mean values, error bars are S.D., $n = 10$ (H1047R) $n = 11$ (N1068fs) or $n = 12$ (WT, M1043L, G1049R.). Experiments were performed with 19.5 to 0.59 nM PI3K, and 100 μM ATP in the absence of lipid vesicles. Two-tailed $t$-test p-values represented by the symbols are as follows: **<0.001; *<0.05; n.s. > 0.05. **C** Protein-Lipid FRET assay performed with different p110α and p85α constructs under basal and pY activated states on PM mimic liposomes containing 5% PIP$_2$, 10% Dansyl PS, 15 % PS, 40% PE, 10% cholesterol, 15% PC and 5% SM. Experiments were carried out at saturating concentrations of PI3K (0.5–1 μM) and 16.65 μg/ml of lipid vesicles (mean is shown, with error bars representing S.D., $n = 3$). Two-tailed $t$-test p-values represented by the symbols are as follows: **<0.001; *<0.05; n.s. > 0.05. Source data for this figure are provided in the Source Data file.

(Supplementary Fig. 5D). H1047R, G1049R and N1068fs all showed significantly increased membrane binding over WT both with and without pY (Fig. 4C), while the M1043L mutant only showed a slight increase in membrane binding without pY for the PM mimic vesicles (Fig. 4C). This clustered the mutants into three groups, H1047R and G1049R which increased basal ATPase and membrane binding, M1043L which increased basal ATPase with only a limited effect on membrane binding, and N1068fs which did not alter basal ATPase, but did increase membrane binding.

### Conformational changes in oncogenic mutations C-terminus of the kinase domain

To test if C-terminal mutations worked by disrupting the inhibitory interaction with the C-terminus, we carried out HDX-MS studies on six constructs of full-length p110α (WT, M1043L, H1047R, G1049R, N1068fs, and a construct lacking the C-terminus (1–1048, referred to afterwards as ΔCter), all bound to full-length p85α. The use of the ΔCter construct allowed for a direct comparison of removal of the C-terminus, versus a possible reorientation upon oncogenic mutation. We used kinase active M1043L, H1047R, G1049R, and a kinase dead variant of N1068fs for this comparison.

HDX-MS experiments were carried out for 4–5 timepoints of exchange (3 s at 1 °C, 3, 30, 300, and 3000 s at 20 °C) for each complex. The full set of all peptides analysed for both p110α and p85α are shown in the Source data, with peptide exchange data presented in Supplementary Fig. 5. The changes observed for H1047R matched almost exactly our previous studies[16], but this experiment expanded the HDX time course, allowing for a more in-depth analysis of perturbations in conformation.

The H1047R, G1049R, and the ΔCter constructs showed similar significant increases compared to the WT in the kinase domain (Fig. 5A–C). These included regions covering 850–858 (hinge between the N and C lobes), the activation loop (930–956) and helices spanning the regulatory arch (1014–1021 in kα10, 1021–1038 in kα11; Fig. 5A–C). Many of these regions correspond directly to the contact site between the C-terminus and kα11. This validates the inhibitory interaction observed in the 4OVU crystal structure, with this interface stabilising helix kα11 and the activation loop. One region in kα8 (962–980) showed a significant change between H1047R, G1049R and WT, which was not observed comparing WT and ΔCter. This helix contacts both the C-terminus and kα11. A possible mechanism explaining this difference is the rotation that occurs in kα11 upon mutation of H1047R.

Structures of free p110α lacking the C-terminus do not show this same rotation of the kα11 helix. The rotation of kα11 would alter the interface between kα8 and kα11, leading to increased exchange in kα8. This could be driven by a unique orientation of the C-terminus packing against kα7 in the H1047R or G1049R mutant, which would be lost in the absence of the C-terminus as would occur in the ΔCter protein. Overall, this data suggests that H1047R and G1049R lead to activation through disruption of the inhibitory conformation of the C-terminal tail, which reorients the lipid binding WIF motif at the membrane binding surface. Both M1043L and N1068fs showed no significant differences compared to WT, suggesting these mutants do not disrupt the inhibitory C-terminal conformation (Fig. 5D + E).

We also compared HDX-MS differences in full-length p110α-p85α between WT, H1047R and ΔC in the presence and absence of pY (Supplementary Fig. 6). The binding of pY led to significant increases for all three constructs at interfaces that have been previously described (ABD-RBD linker, C2-iSH2, nSH2-helical). However, intriguingly there were unique differences upon pY binding for the ΔCter and H1047R constructs at the interface of the regulatory arch with the nSH2 domain (1014–1021, kα10). This is intriguing as it suggests that the opening of this portion of the regulatory motif only occurs upon both disengagement of the nSH2 domain (mediated by pY binding) and disengagement of the inhibitory C-terminus (Supplementary Fig. 6).

## Discussion

Understanding how p110α is regulated and how oncogenic *PIK3CA* mutants alter this regulation is essential in the development of novel PI3K therapeutics. Due to the critical homeostatic roles that WT p110α plays in growth, development and metabolism[40], as well as the cell intrinsic[41,42] and systematic[43] negative feedback loops that oppose PI3K pharmacological inhibition, there are extensive advantages to selectively targeting mutant p110α over WT p110α. Therefore, defining the exact molecular basis underlying mutant specific conformational changes may reveal opportunities for mutant-selective drug design. Oncogenic mutants in p110α span multiple domains of the catalytic subunit, with foci occurring at the ABD, ABD-RBD linker, C2-iSH2 interface, nSH2-helical interface, and the N + C lobes of the kinase domain[44]. While most mis-sense oncogenic mutations discovered in tumours are at hot-spot locations (E542, E545K and H1047R), more than 25% of all mutations in *PIK3CA* occur outside the hotspots. Extensive biochemical[16] and cellular experiments[45,46] have defined that

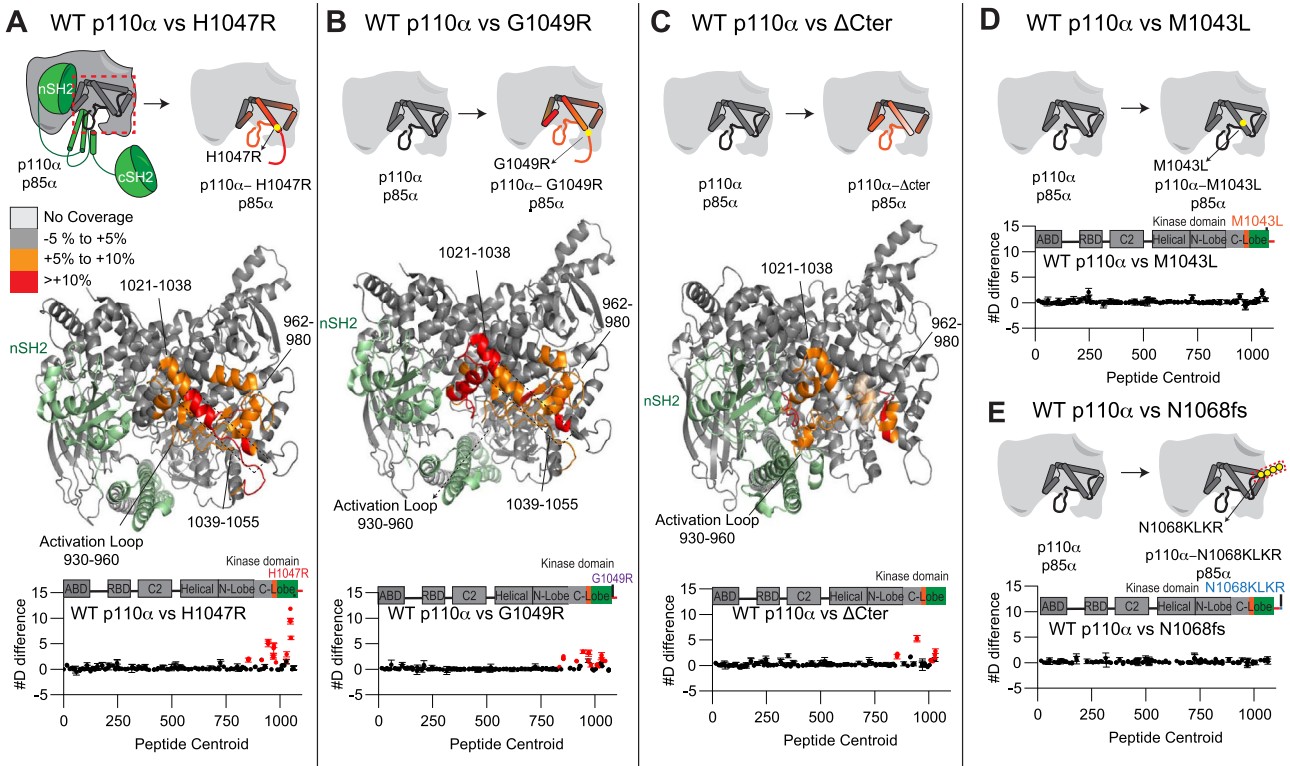

**Fig. 5 | Structural difference between various c-terminal mutants of p110α compared to WT p110α/p85α, and mapping of the p110α membrane binding interface. A–C** HDX comparing p110α/p85α WT vs H1047R (**A**), G1049R (**B**) and ΔCter (1–1048) (**C**). Significant differences in deuterium exchange are mapped on to the structure of p110α/p85α H1047R according to the legend (PDB: 3HHM] (A + B) and 4OVU (C)). The graph of the sum of number of deuteron difference in deuterium incorporation for p110α in each experiment is shown below, with each point representing a single peptide. Peptides coloured in red are those that had a significant change in the mutants (>0.4 Da and 5% difference at any timepoint, with

a two-tailed *t*-test *p* < 0.01). Error bars are shown as the sum of S.D. across all time points. (*n* = 3 for each time point). **D**, **E** HDX comparing p110α/p85α WT vs M1043L (**D**) and N1068fs (**E**). The graph of the #D difference in deuterium incorporation for p110α in each experiment is shown below, with each point representing a single peptide. Error bars are shown as the sum of S.D. across all time points (*n* = 3 for each time point). For all panels, the HDX-MS data for p85 subunits is shown in Supplementary Fig. 5, along with representative peptides showing significant changes. The full HDX-MS data is available in the source data.

*PIK3CA* mutations activate lipid kinase activity by different mechanisms. This together with the recent discovery that multiple mutations in *PIK3CA* occur *in cis*[34], where hotspot and rare mutations are found in the same gene, highlights the need to fully understand how multiple oncogenic mutations can synergically activate PI3K. Here using a suite of biochemical and biophysical approaches we propose a unifying molecular model for how all *PIK3CA* mutations can lead to increased kinase activity, as well as how different oncogenic mutations *in cis* can increase oncogenicity.

We wanted to define how extensive oncogenic gain of function *PIK3CA* mutations at both the ABD interface and the C2-iSH2 interface activate PI3K. The critical role of the ABD in regulating p110α kinase activity was originally suggested by its direct interaction with the N-lobe of the kinase domain[15,47]. We previously observed exposure occurring at the ABD interface (ABD-RBD linker) upon phosphopeptide pY binding[16,19], and more extensive exposure at the ABD interface upon membrane binding[12,16]. Cryo EM analysis of p110α with p85α revealed disengagement of the ABD and p85 subunit from the catalytic core of p110α upon binding phosphorylated peptide[27]. To fully define the relative role of ABD disengagement in PI3K activation on membranes we generated the p110α catalytic core and directly compared its exchange to the full-length p110α-p85α. Increased exposure was seen at all contact interfaces with both p85 and the ABD. Intriguingly, we found that the p110α catalytic core (p110α core) showed H/D exchange rates at the ABD-RBD linker that almost exactly matched those observed upon Ras and pY mediated membrane binding of WT p110α-p85α[12]. Importantly, we did not observe similar rates when

comparing phosphopeptide bound p110α-p85α, suggesting that complete ABD disengagement only occurred when p110α is membrane bound which is in contrast from what was suggested from the cryo-EM model of PI3Kα in the presence of pY peptides.

We previously observed that mutations at the ABD interface exposed the C2-iSH2 interface, and vice versa[16]. As the ABD will always remain tightly bound to the iSH2 domain, its disengagement would likely occur simultaneously with disruption of the C2-iSH2 interface. For these mutants, full disruption of the ABD and C2-iSH2 interfaces occurred only upon removal of the nSH2 (through pY binding), with no requirement for membrane. This reveals an unexpected inhibitory role of the nSH2 domain, whereby packing against the helical, kinase, and C2 domains of p110α and the iSH2 of p85α, it can stabilise the interface with the iSH2 domain, which prevents ABD disengagement from the catalytic core of p110α. Intriguingly, ABD disengagement is likely involved in membrane recruitment for all class IA PI3Ks, as HDX-MS experiments on membranes showed exposure in the iSH2 and the ABD-RBD linker of p110β[48] and p110δ[25,49]. These changes were also seen in primary immunodeficiency mutations of p110δ[50] and in activating *PIK3R1* truncations at the N + C termini of the iSH2 involved in immunodeficiencies and oncogenic transformation[11,49]. An interesting implication in this model is that unique class IA regulatory subunits may have distinctive propensities for ABD-p85 disengagement, which may partially explain a differential role for regulatory subunits in oncogenic transformation[51]. This will require further study to investigate regulatory subunit isoform differences in disengagement. Together this

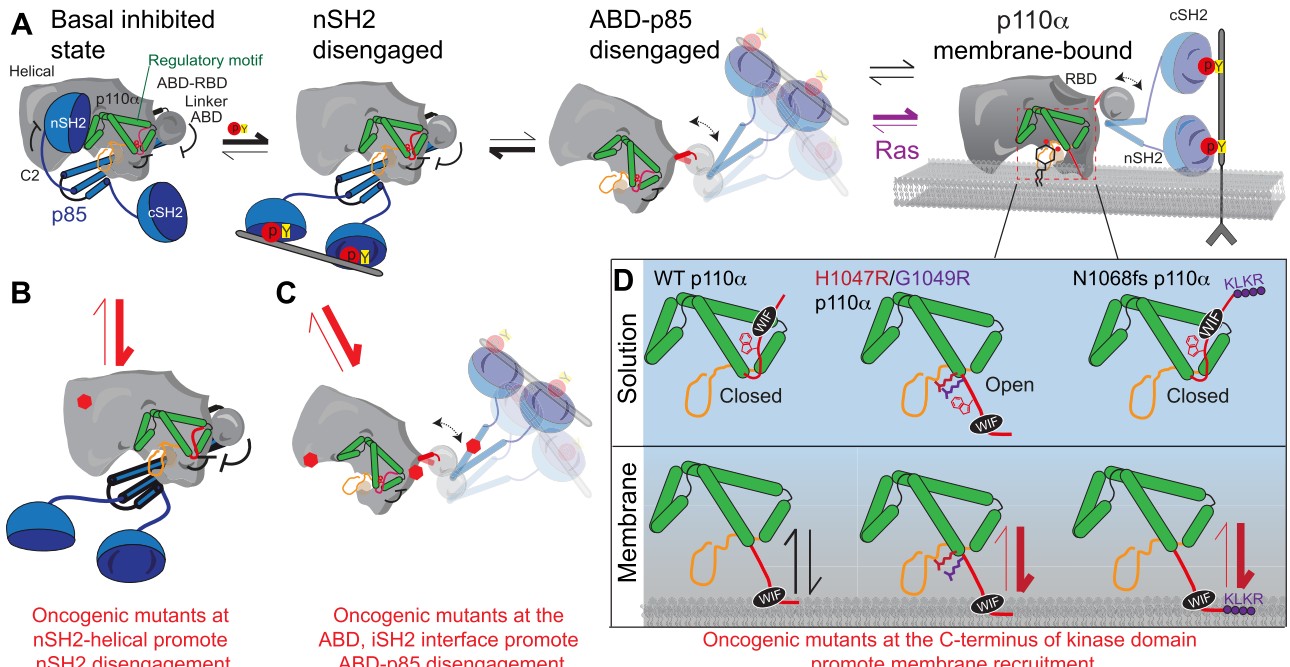

**Fig. 6 | Molecular mechanism of activation of WT PIK3CA, and how oncogenic mutations mimic this process.** Summary of molecular mechanisms of PI3K inhibition by the ABD and regulatory subunit, how activation occurs for wild-type p110α-p85 (**A**), and how oncogenic mutations can alter this process (**B–D**). **A** Proposed mechanism of activation of wild-type p110α-p85α. Activation is initiated by nSH2 disengagement through binding pYXXM motifs (pY), followed by ABD-p85 disengagement, followed by membrane binding (which can be promoted through binding to membrane localised Ras). **B** Activation of p110α-p85α by oncogenic mutants that promote the nSH2 disengagement step of PI3K activation (nSH2-helical hot spot-mutants, E542K, E545K). See Supplementary Fig. 7 for complete list of mutants. **C** Activation of p110α-p85α by oncogenic mutants that promote the ABD-p85 disengagement step of PI3K activation (ABD, ABD-RBD

linker, C2-iSH2 mutants). See Supplementary Fig. 7 for complete list of mutants. **D** Different molecular mechanisms driving activation of C-terminal mutations in p110α. The regulatory motif is coloured green, with the C-terminus coloured red, and the activation loop in black (inactive) or red (orange). The C-terminus when in its closed conformation has the membrane binding WIF motif oriented away from the membrane surface. Membrane binding requires the reorientation of this tail, with the membrane itself likely involved in this conformational change. Mutations that disrupt this interface (H1047R, G1049R) are in an open conformation, leading to greatly increased membrane binding. In the N1068fs mutant there is no change in conformation in solution, but the added KLKR motif dramatically increases membrane recruitment.

highlights the key role of ABD disengagement in PI3K activation, and how oncogenic mutants can alter this regulation.

We next wanted to understand how ABD and p85 disengagement is involved in membrane binding of PI3K. We hypothesised that the free catalytic core of p110α may more efficiently bind to lipid membranes. Our protein-lipid FRET experiments with the p110α core showed enhanced membrane binding compared to the pY activated p110α-p85α heterodimer, which was similar to our previous studies on the catalytic core construct of p110δ[25]. To understand the mechanism for how this occurs we mapped the membrane binding interface of p110α core using HDX-MS. We observed decreased exchange at the C2 and the N-lobe and C-Lobe of the kinase domain. Many of these changes were not observed in the full-length heterodimer upon binding membranes, with these regions located at either the ABD (kα1-kα2 helices of N-lobe) or iSH2 (C2 domain) interfaces. This suggests that some of the membrane binding regions of p110α are shielded by the ABD or p85 subunit, and disengagement of these regions allows for efficient membrane binding. This model also suggests that the role of phosphopeptide binding in membrane recruitment is twofold as it both breaks an inhibitory contact between the nSH2 and the regulatory motif of the kinase domain, and breaks the nSH2 contact with the helical domain, which weakens the interface of the ABD and p85 subunits with the catalytic core, exposing membrane binding surfaces of p110α (Fig. 6A).

One of the largest decreases in exchange observed upon membrane binding was in the C-terminus of p110α, specifically at the membrane binding WIF motif. In high resolution structures of inhibited p110α, the C-terminus is pointed away from the putative

membrane binding surface[36], with the C-terminus pointed towards the membrane surface in the X-ray structure of H1047R p110α[17] due to a ~180 rotation at the end of the kα11 helix. The C-terminus in the WT structures makes a set of contacts the regulatory motif helices (kα8-kα11) located after the activation loop and contacts the C-terminus of the activation loop. The regulatory motif helices and the C-terminus encompass the activation loop and make extensive contacts proposed to maintain an inactive conformation. The regulatory motif is also directly in contact with the nSH2 domain of regulatory subunits. Sequestering the C-terminus in a inhibited conformation is a conserved aspect of regulation across all class I PI3Ks, with the inhibited conformation of the C-terminus of p110β and p110δ binding to the cSH2 of regulatory subunits[25,26], and p110γ having its C-terminus inhibited by an inhibitory Tryptophan lock which does not require regulatory subunits[33]. Removal of the C-terminus causes a substantial rearrangement of the activation loop, and a rotation of the kα11 helix[52]. Most of the frequent mutations in the C-terminus of p110α would be expected to disrupt this inhibitory contact, either through steric hindrance (M1043I/L, H1047R, G1049R) or disruption of key hydrogen bonds (N1044S). One of the only frequent oncogenic mutations that is not at either the regulatory C-terminus or an interface with the ABD or p85 is located at the loop between kα1–kα2 in the N-lobe (E726K) in a location at the membrane binding surface. This mutant likely drives increased lipid kinase activity through enhanced membrane binding of negatively charged lipids through this charge reversal mutant.

To test if conformational changes in H1047R were caused by disruption of the inhibitory contacts with the C-terminus, we compared the H/D exchange rates of H1047R and a deletion of the

C-terminus to the WT protein. The H1047R and C-terminal deletion showed similar increases in exchange in the kα11 helix in contact with the inhibited C-terminus. This suggests a key role of the H1047R mutant is to disrupt inhibitory contacts between the C-terminus and orient the C-terminus in a productive conformation for membrane binding. The H1047R and G1049R, mutants also led to increased activity towards ATP hydrolysis. Therefore, we propose that mutations that disrupt this inhibitory C-terminal contact (H1047R, G1049R) activate PI3K by two unique mechanisms: they reorient the WIF motif towards the membrane binding surface, increasing membrane binding, while also causing the activation loop to adapt a catalytically competent conformation.

There was no increase in ATPase activity for N1068fs compared to wild-type, but it did lead to increased membrane binding. A putative mechanism explaining this is that the 1068 fs replaces the terminal Asn, with a set of positive and hydrophobic residues (KLKR), which all would contribute enhanced membrane binding. So, while the 1068fs mutant does not reorient the membrane binding interface, it likely interacts more extensively, increasing membrane residency time, and increasing PIP$_3$ production. The M1043L mutant has a much smaller effect on membrane binding but did increase basal ATPase rate. Comparing the structure of WT p110α to H1047R p110α shows a rotation of the kα11 helix in the mutant, with the M1043 residue rotating into where the W1051 from the C-terminus is located in the WT structure[17]. We expect that reorientation of the C-terminus is required for full activity, and therefore we propose that the enhanced membrane binding of M1043L (and potentially M1043I/V) is driven by the added steric bulk of the branched chain amino acids, which may more efficiently reorient the C-terminus upon membrane binding. Further structural studies of the M1043 mutants on membrane surfaces will be required to fully access the molecular basis of activation.

Overall, integrating our experimental efforts with the extensive previous biophysical and biochemical analysis of p110α mutants we can propose a unifying model for how p110α is inhibited by regulatory subunits, and how it can be activated both by activating partners and oncogenic mutants (Fig. 6B–D). Activation of p110α/p85α by bis-phosphorylated receptors and their adaptors leads to breaking of the inhibitory nSH2 contacts with the p110α subunit. Helical domain mutants (E542K, E545K) mimic this conformational change (red equilibrium arrows between Fig. 6A-B). The removal of the nSH2 partially destabilises the iSH2-C2 and ABD-kinase interfaces, but in the wild-type complex the equilibrium is mainly towards a closed p85-ABD engaged complex (black arrows). The pYXXM binding surface of the nSH2 exposed in helical mutants leads to greatly enhanced recruitment to RTKs and their adaptors[53]. Oncogenic mutations that occur at the ABD or iSH2 interfaces in p110α (red equilibrium arrows between Fig. 6A–C) shifts the equilibrium towards the disengaged state, leading to increased affinity for membranes. Full membrane recruitment still depends on the opening of the C-terminus of p110α. In the wild-type complex, membrane recruitment requires additional signalling inputs from Ras GTPases. Oncogenic mutants in the kinase domain lead in increased membrane binding either through reorientation of the WIF binding motif and the activation loop (i.e. H1047R, G1049R), or by altering residues at the membrane interface that can more extensively interact with negatively charged membranes (i.e. E726K, 1068 fs) (red equilibrium arrows in Fig. 6D). Together, this model can account for the putative mechanism of activation for >98% of all *PIK3CA* mutants reported in the COSMIC database, and for why double *PIK3CA* mutants lead to increased oncogenicity (Supplementary Fig. 7).

Extensive efforts have focused on the development of p110α selective inhibitors. *PIK3CA* mutant selective inhibitors would likely have major advantages in cancer treatment as they can evade feedback mechanisms that counteract pharmacological inhibition of WT *PIK3CA*. Mutant selective inhibitors have been discovered that lead to selective degradation of E545K and H1047R over WT[54], along with recent reports

of H1047R selective inhibitors. Our data showing very similar conformational changes in H1047R and G1049R suggest that 1047R/L selective small molecules may also be useful in targeting G1049R/S. There also could be advantages of generating tool compounds that activate PI3K activity, with small molecules that disrupt the ABD or p85 interfaces likely acting as activators. Our findings provide a molecular framework for the future development of the next generation of PI3K modulators as therapeutics.

## Methods

### Plasmid generation
Genes of interest were inserted into the pFastBac1 vector to allow baculovirus expression in *Spodoptera frugiperda* (Sf9) cells. The full list of all plasmids and reagents utilised in this manuscript are shown in Table 1. The plasmids containing p110 and free p85α isoforms also expressed N-terminal to the protein a 10X histidine tag, followed by a 2X Strep tag, followed by a Tobacco Etch Virus protease cleavage site. Single substitution mutations (D915N, H1047R, M1043L, G1049R and N1068fs) and p110 truncations (p110α core and ΔC) were generated using site-directed mutagenesis according to published commercial protocols (QuickChange Site-Directed Mutagenesis, Novagen). DNA oligonucleotides spanning the desired region and either containing the altered nucleotides (single substitutions) or lacking the truncated region were ordered (Sigma). PCR reactions were performed on the WT p85α, and PCR purified (Q5 High-Fidelity 2X MasterMix, New England Biosciences #M0492L; QiaQuick PCR Purification Kit, Qiagen #28104). Single colonies were grown overnight and purified using QIAprep Spin Miniprep Kit (Qiagen #27104). Plasmid identity was confirmed by sanger sequencing (Eurofins Genomics and Plasmidsaurus).

### Virus generation and amplification
The plasmids harbouring p110α WT and other variants and p85α were transformed into DH10MultiBac cells (MultiBac, Geneva Biotech) containing the baculovirus viral genome (bacmid) and a helper plasmid expressing transposase to transpose the expression cassette harbouring the gene of interest into the baculovirus genome. Bacmids with successful incorporation of the expression cassette of pFastBac/pACEBac1 into the viral genome was identified by blue-white screening and were purified from a single white colony using a standard isopropanol-ethanol extraction method. Briefly, colonies were grown overnight (~16 h) in 3–5 mL 2xYT (BioBasic #SD7019). Cells were pelleted by centrifugation and the pellet was resuspended in 225 μL P1 Buffer (Qiagen MiniPrep Kit, #27106), chemically lysed by the addition of 225 μL Buffer P2, and the lysis reaction was neutralised by addition of 300 μL Buffer N3. Following centrifugation at 21130 rcf and 4 °C (Rotor #5424 R), the supernatant was separated and mixed with 600 μL isopropanol to precipitate the DNA out of solution. Further centrifugation at the same temperature and speed pelleted the bacmid DNA, which was then washed with 500 μL 70% ethanol three times. The Bacmid DNA pellet was then dried for 1 min and re-suspended in 50 μL Buffer EB.

Purified bacmid was then transfected into Sf9 cells. 2 mL of Sf9 cells between 0.3–0.5 × 10^6 cells/mL were aliquoted into the wells of a six-well plate and allowed to attach, creating a monolayer of cells at ~70–80% confluency. Transfection reactions were prepared by the addition of 2–10 μg of bacmid DNA to 100 μl 1xPBS and 12 μL polyethyleneimine (PEI) at 1 mg/mL (Polyethyleneimine "Max" MW 40,000, Polysciences #24765, USA) to 100 μL 1xPBS. The bacmid-PBS and the PEI-PBS solutions were mixed, and the reaction occurred for 20–30 min before addition drop-by-drop to an Sf9 monolayer containing well. Transfections were allowed to proceed for 5–7 days before harvesting virus containing supernatant as a PI viral stock.

Viral stocks were amplified by adding P1 viral stock to suspension Sf9 cells between 1–2 × 10^6 cells/mL at a 1/100 volume ratio.

## Table 1 | Source and identifier for reagents/resources used

| Reagent or resource | Source | Identifier |
|---|---|---|
| **Bacterial and virus strains** | | |
| *E.coli* XL10-GOLD KanR ultracompetent cells | Agilent | 200317 |
| *E.coli* DH10EMBacY competent cells | Geneva Biotech | DH10EMBacY |
| **Chemicals, peptides, and recombinant proteins** | | |
| H2N-ESDGG(pY)MDMSKDESID(pY)VPMLDMKGDIKYADIE-OH | New England Peptide | N/A |
| Deuterium oxide 99.9% | Sigma Aldrich | 151882-10X1ML |
| ATP | Sigma | A7699- 1g |
| sodium deoxycholate | Sigma | D6750 |
| Protease inhibitor cocktail | Sigma | 535140 |
| Polyoxyethylene (10) lauryl ether | Sigma | P9769 |
| Phosphatidylserine (Porcine Brain) | Avanti | 840032C |
| Phosphatidylethanolamine (Egg yolk) | Sigma Aldrich | P6386 |
| Cholesterol | Sigma Aldrich | 47127-U |
| Phosphatidylcholine (Egg yolk) | Avanti | 840051C |
| Phosphatidylinositol-4,5-bisphosphate (Porcine Brain) | Avanti | 840046 |
| Sphingomyelin (Egg yolk) | Sigma Aldrich | S0756 |
| Dansyl-phosphatidyl serine | Avanti | 810225C |
| 6-well sticky-side chamber | IBIDI | 80608 |
| Polyethanolamine (PEI) | Polysciences | 24765 |
| **Critical commercial assays** | | |
| Transcreener ADP2 FI Assay (1000 Assay, 384 Well), | BellBrook Labs | 3013-1K |
| **Oligonucleotides** | | |
| Forward Primer for D915* mutation in p110α- CCTTCATTTTGGGAATTGGAaatCGTCAC AATAGTAACATCATGGTGAAAGACG | Sigma Aldrich | oHP006F |
| Reverse Primer for D915* mutation in p110α- attTCCAATTCCCAAAATGAAGGTAGCTACACAG | Sigma Aldrich | oHP006R |
| Forward Primer for H1047R* mutation in p110α- CAAATGAATGATGCACgTCATGGTGGCTGG ACAACAAAAATGG | Sigma Aldrich | MR81f |
| Reverse Primer for H1047R mutation in p110α- TGCATCATTCATTTGTTTCATGAAATACTCCAAAGC | Sigma Aldrich | MR81R |
| Forward Primer for M1043L mutation in p110α- GAGTATTTCATGAAACAAcTGAATGATGCACATCAT GGTGGCTGG | Sigma Aldrich | MR82F |
| Reverse Primer for M1043L mutation in p110α- TTGTTTCATGAAATACTCCAAAGCCTCTTGC | Sigma Aldrich | MR82R |
| Forward Primer for G1049R* mutation in p110α- ATGATGCACATCATcGTGGCTGGAC | Sigma Aldrich | oHP005F |
| Reverse Primer for G1049R mutation in p110α- ACgATGATGTGCATCATTCATTTGTTTCATGAAATACTC | Sigma Aldrich | oHP005R |
| Forward Primer for N1068KLKR mutation in p110α- ATGCATTGAAACTTAAAAGGTGAgcggccgctcgagtctag | Sigma Aldrich | oHP004F |
| Reverse Primer for N1068KLKR mutation in p110α- CCTTTTAAGTTTCAATGCATGCTGTTTAATTGTG TGGAAGATC | Sigma Aldrich | oHP004R |
| Forward Primer for ΔC mutation in p110α- GGCgccggtaccATGCCTCCACGACCATCATCAGG | Sigma Aldrich | MR80F |
| Reverse Primer for ΔC mutation in p110α- agcggccgcTCAATGATGTGCATCATTCATTTGTTTCATGAAATAC | Sigma Aldrich | MR86R |
| Forward Primer for p110α core mutation in p110α- GGGCgccggtaccAACCGTGAAGAAAAGATCCTCAATCGAG | Sigma Aldrich | oHP008F |
| Reverse Primer for p110α core mutation in p110α- GGATCTTTTCTTCACGGTTggtaccggcGCCCTGAAAATAC | Sigma Aldrich | oHP008R |
| **Recombinant DNA** | | |
| Human p110α (1-1068) in pACEBac1 vector | Dornan et al.[11] | GD102 |
| p110α H1047R in pACEBac1 vector | This paper | MR98 |
| p110α M1043L in pACEBac1 vector | This paper | MR100 |
| p110α G1049R in pACEBac1 vector | This paper | HP15 |
| p110α with N1068KLKR in pACEBac1 vector | This paper | HP14 |
| p110α ΔCter (1–1048) in pACEBac1 vector | This paper | MR106 |
| p110α core (106–1068) in pACEBac1 vector | This paper | HP31 |
| Human p85α (1-724) | Dornan et al.[11] | EX20 |
| p110α D915N (1-1068) in pACEBac1 vector | This paper | HP16 |
| p110α D915N and H1047R in pACEBac1 vector | This paper | HP19 |
| p110α D915 and M1043L in pACEBac1 vector | This paper | HP25 |
| p110α D915 and G1049R in pACEBac1 vector | This paper | HP27 |
| p110α D915 and N1068KLKR in pACEBac1 vector | This paper | HP26 |
| Free p85α in pFastBac 1 | This paper | AS01 |

**Table 1 (continued)**

| Reagent or resource | Source | Identifier |
|---|---|---|
| Software and algorithms | | |
| HDExaminer | Sierra Analytics | http://massspec.com/hdexaminer |
| GraphPad Prism 7 | GraphPad | https://www.graphpad.com |
| Adobe Illustrator | Adobe | https://www.adobe.com/products/illustrator.html |
| COSMIC: Catalogue Of Somatic Mutations In Cancer | Sanger; | https://cancer.sanger.ac.uk/cosmic |
| PyMOL | Schroedinger | http://pymol.org |
| Other | | |
| Sf9 insect cells for expression | Expression Systems | 94-001S |

This amplification produces a P2 stage viral stock that can be used in final protein expression. The amplification proceeded for 4–5 days before harvesting, with cell shaking at 120 RPM in a 27 °C shaker (New Brunswick). Harvesting of P2 viral stocks was carried out by centrifuging cell suspensions in 50 mL Falcon tubes at 2281 RCF (Beckman GS-15), collecting the supernatant in a fresh sterile tube, and adding 5–10% inactivated foetal bovine serum (FBS; VWR Canada #97068-085).

### Expression and purification of recombinant proteins

All PI3K constructs were purified by expressing the catalytic subunit and the regulatory subunit using the pFASTBAC/ pACEBac1 expression system in Sf9 cells. After expressing the cells at 27 °C (42 h for Kinase active variants and 55 h for Kinase dead), the cells were harvested at 1739 × $g$ at 4 °C using Eppendorf Centrifuge 5810R and the cells were flash frozen using liquid nitrogen and stored in −80 °C.

The frozen pellets were resuspended in lysis buffer containing 20 mM Tris pH 8, 10 mM Imidazole, 100 mM NaCl, 5% glycerol [v/v], 2 mM βME, protease inhibitor [Protease Inhibitor Cocktail Set III, Sigma]) and sonicated for 2 min (15 s on, 15 s off, level 4.0, Misonix sonicator 3000). Triton-X 100 was added to the lysate at a final concentration of 0.1% and then clarified by spinning at 15,366 × $g$ for 45 min (Beckman Coulter JA-20 rotor). The supernatant was loaded onto a 5 ml crude Ni-NTA column (GE healthcare) equilibrated in NiNTA A buffer containing 20 mM Tris pH 8, 100 mM NaCl, 10 mM Imidazole and 5% glycerol [v/v]. The column was washed using high salt buffer containing 20 mM Tris, 1 M NaCl, 10 mM Imidazole, 5% Glycerol [v/v] followed by NiNTA buffer wash (20 mM Tris pH 8, 100 mM NaCl, 21 mM Imidazole and 5% Glycerol). The protein was eluted using 100% NiNTA B buffer (20 mM Tris pH 8, 100 mM NaCl, 200 mM Imidazole and 5% Glycerol). The elute from the nickel column was loaded onto Streptavidin column (GE healthcare) and subjected to buffer wash using Hep A buffer (20 mM Tris pH 8, 100 mM NaCl, 5% Glycerol and 0.5 mM tris(2-carboxyethyl) phosphine [TCEP]). The column was incubated on ice for 3 h in the presence of TEV protease and then eluted by a wash with HEP A buffer. The eluent was loaded onto Q column equilibrated with HEP A buffer. The column was washed with HEP A buffer to remove TEV protease and the final PI3K was eluted by passing Hep B elution buffer (20 mM Tris pH 8, 325 mM NaCl, 5% glycerol [v/v] and 0.5 mM TCEP). The protein was exchanged to the final buffer containing 20 mM HEPES pH 7.5, 100 mM NaCl, 10% Glycerol [v/v] and 0.5 mM TCEP using a desalting column. The protein was concentrated to 1 mg/ml using a 50,000 MWCO Amicon Concentrator (Millipore), flash frozen and stored at −80 °C.

For HDX experiments involving WT, p110α core, H1047R, M1043L, ΔCter and free p85α were further subjected to gel filtration using Superdex™ 200 10/300 GL Increase from GE healthcare. After gel filtration, the protein was concentrated, aliquoted, frozen and stored at −80 °C.

### Lipid vesicle preparation

To measure membrane recruitment using Protein-Lipid FRET two sets of lipid vesicles were prepared: PE/PS/PIP$_2$ containing 5 % brain PIP$_2$, 65% egg yolk PE, 25 % brain PS and 10% Dansyl-PS (Avanti, #810225 C) and PM mimic consisting of 5% brain PIP$_2$, 20% brain PS, 10% Dansyl-PS, 45% egg yolk PE, 15% egg yolk phosphatidylcholine (PC) (Avanti #840051C), 10% cholesterol (Sigma Aldrich, #47127-U) and 5% egg yolk sphingomyelin (Sigma Aldrich, #S0756). To generate vesicles the lipid mixtures were combined in organic solvent. The mixture was then evaporated using a stream of argon gas followed by desiccation under vacuum for 45 min. The lipids were resuspended in a lipid buffer (25 mM HEPES pH 7, 100 mM NaCl, 10% Glycerol [v/v]) and the solution was subjected to sonication for 15 mins. The vesicles were subjected to five freeze thaw cycles and extruded 11 times through a 100 nm filter (T and T Scientific: TT-002–0010). The extruded vesicles were sonicated again for 5 min, aliquoted and stored at −80 °C.

### Protein−lipid FRET assay

Protein-lipid FRET experiments were carried out either at a saturating protein concentration (Fig. 4C, also Supplementary Fig. 4E, F) or as a dose response with PI3K (Supplementary Fig. 3B). Protein lipid FRET assays with saturating PI3K were initiated by mixing 2.5 µL PI3K (final concentration of 1 µM) with 2.5 µL of pY peptide diluted in protein buffer [20 mM HEPES pH 7.5, 100 mM NaCl, 10 % Glycerol [v/v]] (final concentration of 1 µM) for 15 mins at 20 °C. 5 µL of lipid vesicles (either PIP$_2$/PS/PE or PM mimic vesicles, both at final conc. 16.65 µg/mL) was added to the protein-pY mixture and were incubated for 15 mins at 20 °C. Dose experiment was carried out exactly the same, except PI3K amount varied from 0.015 µM to 1 µM. The plate was then read using a SpectraMax M5 plate reader using a 280-nm excitation filter with 350-nm and 520- nm emission filters to measure Trp and Dansyl-PS FRET emissions, respectively. The FRET signal shown in the figure has I-I$_o$ along the Y axis where I is the intensity of 520 with protein and Io is the intensity of lipid alone.

### ATPase assay

All ATPase assays used the Transcreener ADP2 Fluorescence Intensity (FI) assay (Bellbrook labs) which measures formation of ADP. In all, 2 µL of a kinase solution (final concentration 200nM-0.186 nM) at 2X final concentration was mixed with 2 µL substrate solution containing ATP, and the reaction was allowed to proceed for 60 min at 37 °C. The reaction was stopped with 4 µL of 2X stop and detect solution containing EDTA (chelates Mg$^{2+}$, stopping kinase activity) along with 8 nM ADP Alexa Fluor 594 Tracer and 93.7 µg/mL ADP2 Antibody IRDye QC-1 (Bellbrooks lab kit, 3013-1 K), and was allowed to incubate for 60 min. The fluorescence intensity was measured using a SpectraMax M5 plate reader at excitation 590 nm and emission 620 nm. This data was normalised against a 0–100% ADP window made using conditions

containing a final concentration of 100 µM ATP or ADP. % ATP turnover was interpolated from an ATP standard curve obtained from performing the assay on 100 µM (total) ATP/ADP mixtures with increasing concentrations of ADP.

### HDX-MS analysis: sample preparation

HDX-MS experiments for all conditions were conducted as follows: reactions comparing full-length p110α/p85α and p110α core were conducted in 13.5-µl reaction volumes with a final PI3K amount of 10 pmol. Prior to HD exchange, 3.5 µl of either protein was incubated with 3.5 µl of PIP$_2$/PS/PE lipid (5 % brain PIP$_2$, 65% egg yolk PE, 30 % brain PS) at 2.5 mg/ml final or lipid buffer for 2 min at room temperature. To initiate HD exchange, a mixture of either 6.5 µl of the same lipid at 0.76 mg/ml final or lipid buffer and 36.5 µl of D$_2$O buffer [20 mM HEPES (pH 7.5), 100 mM NaCl, 94.3% D$_2$O (v/v)] was added to the protein-lipid/buffer mix (final D$_2$O concentration of 69%). Exchange was carried out for 3, 30 and 300 s at 20 °C.

HDX reactions comparing full-length p110α/p85α WT against M1043L, H1047R and ΔC were conducted in 20 µl reaction volumes with a final PI3K amount of 9 pmol. Prior to HD exchange, 3 µl of protein was incubated with 1 µl of 50 µM pY peptide or protein buffer and allowed to incubate for 15 min on ice. Exchange was initiated by the addition of 16 µl of D$_2$O buffer to the protein + /– pY mixture (final D$_2$O concentration of 75%). Exchange was carried out for 3, 30, 300 and 3000 s at 20 °C and 0.3 s (3 s on ice).

HDX reactions comparing full-length p110α/p85α WT against G1049R were conducted in 20 µl reaction volumes with a final PI3K amount of 15 pmol. HD exchange was initiated by the addition of 16 µl of D$_2$O buffer to 4 µl of protein (final D$_2$O concentration of 75%). The reaction proceeded for 3, 30, 300 and 3000 s at 20 °C.

HDX reactions comparing full-length p110α/p85α WT against N1068fs were conducted in 20 µl reaction volumes with a final PI3K amount of 11 pmol. HD exchange was initiated by the addition of 17.2 µl of D$_2$O buffer to 2.8 µl of protein (final D$_2$O concentration of 81%). The reaction proceeded for 3, 30, 300 and 3000 s at 20 °C.

HDX reactions comparing full-length p110α/p85α WT kinase active vs kinase dead were conducted in 50 µl reaction volumes with a final PI3K amount of 10 pmol. Protein was then incubated with 1 µM pY peptide and allowed to incubate for 15 min on ice. Prior to HD exchange, 4 µl of either protein was incubated with 10 µl of PIP$_2$/PS/PE lipid (5% brain PIP$_2$, 65% egg yolk PE, 30 % brain PS) at 1 mg/ml or lipid buffer for 2 min at 20 °C. HD exchange was initiated by the addition of 36 µl of D$_2$O buffer (final D$_2$O concentration of 69%). The reaction proceeded for 3, and 300 s at 20 °C. All conditions and timepoints were generated in independent triplicate. All exchange reactions were terminated by the addition of ice-cold quench buffer to give a final concentration 0.6 M guanidine-HCl and 0.9% formic acid. Samples were flash frozen in liquid nitrogen immediately after quenching and stored at −80 °C until injected onto the ultra-performance liquid chromatography (UPLC) system for proteolytic cleavage, peptide separation, and injection onto a QTOF for mass analysis, described below.

### HDX-MS analysis: protein digestion and tandem MS data collection

Protein samples were rapidly thawed and injected onto an integrated fluidics system containing an HDx-3 PAL liquid handling robot and climate-controlled (2 °C) chromatography system (LEAP Technologies), a Dionex Ultimate 3000 UHPLC system, and an Impact HD QTOF mass spectrometer (Bruker). The protein was run over one immobilised pepsin column (Trajan; ProDx protease column, 2.1 mm × 30 mm PDX.PP01-F32) at 200 µl/min for 3 min at 10 °C. The resulting peptides were collected and desalted on a C18 trap column [Acquity UPLC BEH C18 1.7 mm column (2.1 × 5 mm); Waters 186003975]. The trap was subsequently eluted in line with an ACQUITY 1.7 µm particle,

100 × 1 mm$^2$ C18 UPLC column (Waters 186002352), using a gradient of 3–35% B (buffer A, 0.1% formic acid; buffer B, 100% acetonitrile) over 11 min immediately followed by a gradient of 35–80% B over 5 min. MS experiments acquired over a mass range from 150 to 2200 mass/charge ratio (m/z) using an electrospray ionisation source operated at a temperature of 200 °C and a spray voltage of 4.5 kV.

### HDX-MS analysis: peptide identification

Peptides were identified from the nondeuterated samples of p110α/p85α complex for WT and other mutants using data-dependent acquisition following tandem MS (MS/MS) experiments (0.5-s precursor scan from 150 to 2000 m/z: 12 0.25 s fragment scans from 150 to 2000 m/z). MS/MS datasets were analysed using PEAKS7 (PEAKS), and peptide identification was carried out by using a false discovery–based approach, with a threshold set to 1% using a database of purified proteins and known contaminants found in SF9 cells[55]. The search parameters were set with a precursor tolerance of 20 parts per million, fragment mass error 0.02 Da, and charge states from 1 to 8.

### HDX-MS analysis: mass analysis of peptide centroids and measurement of deuterium incorporation

HD-Examiner Software (Sierra Analytics) was used to automatically calculate the level of deuterium incorporation into each peptide. All peptides were manually inspected for correct charge state, correct retention time, and appropriate selection of isotopic distribution. Deuteration levels were calculated using the centroid of the experimental isotope clusters. HDX-MS results are presented with no correction for back exchange shown in the Source data, with the only correction being applied correcting for the deuterium oxide percentage of the buffer used in the exchange (69% for p110α core experiments, 69% for WT vs M1043L, H1047R and ΔC, 75% for WT vs G1049R, 81% for WT vs N1068fs and 69% for WT kinase active vs kinase dead). Attempts to generate a fully deuterated class I PI3K sample were unsuccessful, which is common for large macromolecular complexes. Therefore, all deuterium exchange values are relative.

Differences in exchange in a peptide were considered significant if they met all three of the following criteria: ≥5% change in exchange, ≥0.4 Da difference in exchange, and a $P$-value of <0.01 using a two-tailed Student's $t$ test. The raw HDX data are shown in two different formats.

The raw data for all analysed peptides is available in the source data. The differences in deuterium exchange are visualised in different ways. To allow for visualisation of differences across conditions, we used number of deuteron difference (#D) plots (Figs. 2C, 3D, 5A–E, Supplemental Fig. 3C–E, Supplemental Fig. 4A + C, Supplemental Fig. 5A, and Supplemental Fig. 6D–F). These plots show the total difference in deuterium incorporation over the entire HDX time course, with each point indicating a single peptide. These graphs are calculated by summing the differences at every time point for each peptide and propagating the error. For a selection of peptides, we are showing the %D incorporation over a time course, which allows for comparison of multiple conditions at the same time for a given region (Figs. 2D + 3E, Supplemental Fig. 3F, Supplemental Fig.4B + D, Supplemental Fig.5B + C and Supplemental Fig.6G). Samples were only compared when they were set at the same time and were never compared to experiments completed with a different final D$_2$O level. The data analysis statistics for all HDX-MS experiments are in Supplemental Table 1a–d according to published guidelines. The HDX-MS proteomics data generated in this study have been deposited to the ProteomeXchange Consortium via the PRIDE partner repository[56] with the dataset identifier PXD031080.

### Reporting summary

Further information on research design is available in the Nature Portfolio Reporting Summary linked to this article.

## Data availability

The data that support this study are available from the corresponding authors upon reasonable request. The HDX-MS proteomics data generated in this study have been deposited to the ProteomeXchange Consortium via the PRIDE partner repository[56] with the dataset identifier PXD031080. All data generated or analysed during this study are included in the Source Data file. Structures used are available at the following accession codes: 4OVU [https://doi.org/10.2210/pdb7VF9/pdb] and 3HHM [https://doi.org/10.2210/pdb7VF9/pdb]. Source data are provided with this paper.

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

## Acknowledgements
J.E.B. is supported by the Cancer Research Society (CRS, 843232), and the Michael Smith Foundation for Health Research (MSFHR, scholar 17686).

## Author contributions
M.L.J., H.R.P. and N.J.H. carried out all biochemical analysis in the manuscript. M.L.J., H.R.P., M.A.H.P., M.K.R. and J.E.B. carried out and analysed all HDX-MS data. M.L.J., H.R.P. and J.E.B. designed the experiments. All authors contributed to writing and editing.

## Competing interests
J.E.B. reports personal fees from Olema Pharmaceuticals (San Francisco, USA) and Scorpion Therapeutics (Boston, USA). The remaining authors declare no competing interests.
