## [Peer Review File · Nature Communications]

Oncogenic mutations of PIK3CA lead to increased membrane recruitment driven by reorientation of the ABD, p85 and C-terminusReviewers' Comments:

Reviewer #1:

Remarks to the Author:

This manuscript from Ranga-Prasad et al. dissects how PIK3CA mutations alter membrane binding and recruitment and the conformation of other domains of p110 α and p85 α to activate PI3K. Using HDX-MS in concert with PI3K biochemistry experiments, the authors find that upon membrane binding, 1) the p110 α ABD is dissociated from the catalytic core 2) the p110 α C2 domain is dissociated from the p85 α iSH2 domain, and 3) the p110 α C-terminus is reoriented. They also use a series of C-terminal p110 α mutants to support their findings of allosteric activation mechanisms.

This work is a tour de force in the PI3K biochemistry and structural field, joining several important papers in the field by the Roger Williams, Mario Amzel, and Sandra Gabelli groups, and certainly deserves publication in Nature Communications. This work unifies the PIK3CA variant field, providing a mechanistic explanation for PI3K activation by 98% of PIK3CA cancer associated variants, and will undoubtedly inform mutant-specific PI3K inhibitor drug development. The authors' experiments are rigorous, measuring deuterium exchange with and without pY, membrane, and membrane-bound Ras. Moreover, the complex conformational change data are presented beautifully and easily for the reader.

I have no major comments and several minor comments regarding the text below.

Abstract

Line 27 – Would change "undefined" as we do have some mechanistic insight into the most frequent PIK3CA mutations

Line 32 – "of the p110 α C-terminus"

Line 36 – doesn't make sense, missing "how"?

Introduction

Line 60 – as above, too strong to say they are unknown

Line 129 – should be "in controlling PI3K enzyme activity"

Line 173 – For high affinity interaction between ABD and iSH2 please provide a reference

Line 200 – should be "disengagement of the ABD and the regulatory..."

Line 310 – Figure 4C is mentioned after 4D+E

Line 313 – "Where assays on PIP2 membrane substrates showed similar activation for all mutants compared to WT" should be rephrased – all mutants showed a statistical increase with pY binding, but the increase in H1047R and G1049R was much greater even with a large increase in basal ATPase activity to WT.

Figure 1

A – Domains of p85 should to be connected in schematic

B – Please indicate >50 and >2000 descriptor in figure (mutations, variants, etc.). Also E542 and E545 should be colored red.

C – Would change Y-axis to "number of substitutions/variants"

Figure 4

D,E – It seems like some of the data points do not have error bars. Could the authors please clarify?

Reviewer #2:

Remarks to the Author:

The manuscript by Ranga-Prasad et al. describes the structural effects of oncogenic mutations in PIK3CA that lead to increased membrane recruitment, more specifically by studying the effect of these mutations on p110 α deuterium uptake.

Novel molecular aspects are nicely presented and supported by the data, however some parts of the results are based on previously published data and therefore lack originality. Slight restructuring i.e. putting emphasis only on unpublished data, while mentioning previous ones only when necessary, could help improve the manuscript.

Below some specific questions and comments:

-Results part 1, no title:

Many of the conclusions are based on already published data, and I am not sure about the extent of contribution of the current HDX experiments with the Δ ABD to the final conclusion of this part "Overall, this data supports that disengagement of the ABD and p85 subunit from the p110 α catalytic core occurs upon membrane binding".

Is the increased deuterium uptake of the linker peptide in Δ ABD structurally similar to the increase observed for the same region in the full-length protein bound to p85, RAS and the membranes (Figure 2A and B)? the linker in the Δ ABD becomes an N-terminal unstructured peptide, which can explain the increased deuterium uptake. Could the authors please comment on that?

-Enhanced membrane binding of Δ ABD:

i) Membrane binding by FRET: did the author perform a control test with p85 alone in the presence of the membranes? I am curious about the contribution of the presence of p85 to the FRET signal quantified in these experiments.

ii) Is the enhanced membrane binding only linked to the removal of the ABD domain or to both ABD domain and p85? In other words, was the implication of p85 to the WT p110 α membrane binding assessed?

-Membrane binding surface of Δ ABD:

while the results at the N- and C-lobe are clearly convincing, I am not quite sure about the slight HDX difference in one of the identified peptides in the C2 region. Further experiments (for instance, effect of a critical point mutation in this region on binding to p85) could consolidate the drawn conclusion related to the impact of the C2 domain and therefore the disengagement of p85 in membrane binding.

-Role of the conformation of the C-terminus in PI3K activation:

This part does not provide new experimental data but rather a new analysis of already published data.

-Conformational changes in oncogenic mutations C-terminus:

Impact of the Δ Cter and H1047 mutations in the 1014-1021 peptide are very subtle compared to the observed differences in the rest of the p110 α sequence. In addition, the differences are observed mainly for the longest incubation timepoint, as opposed to the other peptides shown in figure S7. Could the authors comment on that? How comparable are the differences observed in this region and the other ones showing a more obvious deuteration difference (e.g. in the C2 region)?

-In figure S3:

i) Sequence coverage of p85A: Why are there many missing peptides (mainly in the SH2 in p85) in the presence of the membrane?

ii) Why only 2 deuteration timepoints are shown for the different peptides in panel F?

-Table S1: how come number of peptides/length and redundancy are the same between WT and Δ ABD PI3K?

-Please add the state info (State A – State B =?) in all figures showing the #D Difference.

-Data present in figure S4 related to main figure 2B is previously published? If so, please specify in figure legend.

-Overall, some of the main figures are densely packed (like for instance panels C to E in figure 3); consider revising them to make them clearer.

-Below some suggestions and questions for Figure 1:

i) Increase the size of the scheme in panel A.

ii) Is the regulatory motif the part coloured in green or inside the dashed box? Better stick to colouring the regulatory motif and removing the dashed box.

iii) What do the green circles refer to in Panel B structure? consider changing the green colour since it's the colour code chosen for the regulatory motif (unless It's related to that motif).

iv) What does the red arrow relate to in panel E?

Additional information related to the questions above could be added in the figure legend for more clarity.

v) Use the same magenta colour used in the mapped structures for the C-terminus in the schematic in panel A.

REVIEWER COMMENTS

Reviewer #1 (Remarks to the Author):

This manuscript from Ranga-Prasad et al. dissects how PIK3CA mutations alter membrane binding and recruitment and the conformation of other domains of p110 α and p85 α to activate PI3K. Using HDX-MS in concert with PI3K biochemistry experiments, the authors find that upon membrane binding, 1) the p110 α ABD is dissociated from the catalytic core 2) the p110 α C2 domain is dissociated from the p85 α iSH2 domain, and 3) the p110 α C-terminus is reoriented. They also use a series of C-terminal p110 α mutants to support their findings of allosteric activation mechanisms.

This work is a tour de force in the PI3K biochemistry and structural field, joining several important papers in the field by the Roger Williams, Mario Amzel, and Sandra Gabelli groups, and certainly deserves publication in Nature Communications. This work unifies the PIK3CA variant field, providing a mechanistic explanation for PI3K activation by 98% of PIK3CA cancer associated variants, and will undoubtedly inform mutant-specific PI3K inhibitor drug development. The authors' experiments are rigorous, measuring deuterium exchange with and without pY, membrane, and membrane-bound Ras. Moreover, the complex conformational change data are presented beautifully and easily for the reader.

We appreciate the positive assessment of our work and appreciate the comment on this being a tour de force of PI3K biochemistry.

I have no major comments and several minor comments regarding the text below.

Abstract

Line 27 – Would change "undefined" as we do have some mechanistic insight into the most frequent PIK3CA mutations

We agree, and our point here is that we don't understand how all mutants activate activity. We have changed to '*The full set of mechanisms underlying how PI3Ks are activated by all oncogenic mutations on membranes are unclear.*'

Line 32 – "of the p110a C-terminus"

We have corrected this typo

Line 36 – doesn't make sense, missing "how"?

Agreed, revised to

'This work reveals unique mechanisms underlying how PI3K is activated by oncogenic mutations, and explains how double mutants can synergistically increase PI3K activity.'

Introduction

Line 60 – as above, too strong to say they are unknown

Agreed have changed to '*However, the full molecular mechanisms underpinning activation and membrane binding of class IA PI3K, and how oncogenic mutants alter this are not completely understood.*'

Line 129 – should be “in controlling PI3K enzyme activity”

Agreed, corrected to

‘To investigate the role of the ABD domain / p85 regulatory subunit in controlling PI3K enzyme activity, we needed a construct that allowed us to interrogate the dynamic effects of full ABD disengagement.’

Line 173 – For high affinity interaction between ABD and iSH2 please provide a reference
We have included a citation to one of the original purification papers on the p110-p85 complex that reported that no buffer conditions have been identified that are able to disrupt this complex. Due to the instability of the full length p110 subunit it has been impossible to determine quantitative measurements of binding affinity.

34. Fry, M. J. et al. Purification and characterization of a phosphatidylinositol 3-kinase complex from bovine brain by using phosphopeptide affinity columns. Biochemical Journal 288, 383–393 (1992).

Line 200 – should be “disengagement of the ABD and the regulatory...”

Agreed, correction made as follows

Our hypothesis that disengagement of the ABD and the regulatory subunit p85 α subunit is required for membrane binding suggested that there should be differential membrane binding of the p110 α core compared to full length p110 α /p85 α .

Line 310 – Figure 4C is mentioned after 4D+E

We have completely changed this figure as described below, its reference has been fixed in the text.

Line 313 – “Where assays on PIP₂ membrane substrates showed similar activation for all mutants compared to WT” should be rephrased – all mutants showed a statistical increase with pY binding, but the increase in H1047R and G1049R was much greater even with a large increase in basal ATPase activity to WT.

We believe this might have been slightly confusing in the original wording. We believe the reviewer is looking at the ATPase data in Fig 4C (no PIP₂ present).

Due to reasons described below we have removed data generated using the PIP₂ assay, and have rephrased this section to only focus on the ATPase data.

Figure 1

A – Domains of p85 should to be connected in schematic

We have added the connecting line for the domain schematics.

B – Please indicate >50 and >2000 descriptor in figure (mutations, variants, etc.). Also, E542 and E545 should be colored red

We have more clearly described this in the figure and in the figure legend. We have also made sure all mutants that are greater than 2000 are colored red (i.e. 542 and 545). The 545 and 542 lie underneath the transparent blue p85, which may make their color look slightly different, we have described this more clearly in the figure legend.

C – Would change Y-axis to “number of substitutions/variants”

We agree and have changed this as suggested

Figure 4

D,E – It seems like some of the data points do not have error bars. Could the authors please clarify?

These assays were carried out using a PIP₂ binding, with substrate depletion leading to a decrease in signal as the PH domain is removed from the membrane surface. At the reviewers suggestion we re-analyzed the PIP₂ data, and found that a few of these curves were done in singlicate (apologies for this oversight).

We replicated these assays with a new batch of the PIP₂ binding domain and found that the assay performance was highly variable depending on the prep of PIP₂ sensor, limiting its application for an endpoint assay. We have spent the following months attempting to optimise this assay, but have found that this is likely going to be extremely challenging to use as an end-point assay. We then attempted to use the commercially available PI 3-Kinase 3-step HTRF[®] Assay available from Sigma, which worked well with soluble C8 PIP₂ substrate, but had major limitations when used with membrane substrate. It appears that the only suitable approach for carrying out careful analysis requires the use of a P₃₂ radioactive assay, measuring incorporation of radioactive phosphate into PIP₃, which we do not have access to.

For this reason we have removed the lipid kinase data from the manuscript, focusing primarily on the ATPase alone and membrane binding data. Combined with the HDX-MS data, we still believe this defines a clear molecular mechanism of how these mutants are activated. We apologise to the reviewers for this.

Reviewer #2 (Remarks to the Author):

The manuscript by Ranga-Prasad et al. describes the structural effects of oncogenic mutations in PIK3CA that lead to increased membrane recruitment, more specifically by studying the effect of these mutations on p110 α deuterium uptake.

Novel molecular aspects are nicely presented and supported by the data, however some parts of the results are based on previously published data and therefore lack originality. Slight restructuring i.e. putting emphasis only on unpublished data, while mentioning previous ones only when necessary, could help improve the manuscript.

We appreciate the positive assessment of our work and agree with the reviewer that more clearly highlighting the unpublished work would strengthen the manuscript. We have made multiple changes to highlight this, while only referring to previous data when necessary. The only previous data currently shown is in Fig. 2B, 2D, with us removing the previously published data that was originally in Fig 3D in the original manuscript. Individual changes are described below

Below some specific questions and comments:

-Results part 1, no title:

Many of the conclusions are based on already published data, and I am not sure about the extent of contribution of the current HDX experiments with the Δ ABD to the final conclusion of this part “Overall, this data supports that disengagement of the ABD and p85 subunit from the p110 α catalytic core occurs upon membrane binding”.

Is the increased deuterium uptake of the linker peptide in Δ ABD structurally similar to the increase observed for the same region in the full-length protein bound to p85, RAS and the membranes (Figure 2A and B)? the linker in the Δ ABD becomes an N-terminal unstructured peptide, which can explain the increased deuterium uptake. Could the authors please comment on that?

Our conclusion on the importance of ABD disengagement upon membrane binding is based on similar exchange rates for both the catalytic core and the fully activated full length protein occurring in the ABD-RBD linker peptide (114-119, see Fig 2 panel D) that is in contact with the ABD in the full length inactive p110-p85 protein complex. A critical piece of information present in the results in this manuscript is establishing the exchange rate for the free catalytic core, as the 114-119 linker peptide is not unstructured (see Fig 2 panel D, with almost no exchange at 3 seconds), implying the helix is still formed, however it is more dynamic than with the ABD present. This baseline is critical to interpret the membrane bound changes (previously published data, Fig 2D).

With an HDX difference it is hard to define if the change is due to the formation of exactly the same structure. Changes in exchange are similar between the catalytic core and membrane bound PI3K, with the catalytic core being more exposed than even the fully activated. However, we find that the exposure of the linker in the full-length complex is correlated to conditions that increase membrane binding (i.e., partial exposure upon pY mediated membrane binding, and additional exposure upon pY and Ras mediated membrane binding, which drives increased membrane occupancy).

We believe this is linked to the occupancy of the PI3K bound to membrane, and as membrane binding increases it approaches a limit of exposure that is like the free catalytic core. This is supported by our previous studies examining membrane binding of oncogenic mutants at this interface (G106V, G118D, N345K) where we saw enhanced exposure of this region compared to WT upon membrane binding, likely due to the increased membrane binding of oncogenic mutants.

We have added additional details in the results, describing how enhanced membrane binding leads to increased exposure of this linker region, and how this relates to our previous HDX-MS data (see new paragraph below):

"This data comparing the full-length heterodimer vs p110 α core allowed us to define the effect of ABD removal on the contact site at the ABD-RBD linker. This region still is protected from exchange at early time points, suggesting presence of secondary structure, however, it is much more dynamic in the absence of the ABD. Comparing this to previous HDX-MS experiments examining pY-Ras membrane recruitment of p110 α -p85 α ¹², showed that the exchange rate of the core is similar to the p110 α -p85 α membrane bound state, suggesting a correlative ABD disengagement occurring with membrane binding. This is supported by our previous observation of increased membrane binding for oncogenic mutants at the C2-iSH2 or ABD interfaces (N345K, G106V and G118D) that would be expected to promote ABD / iSH2 disengagement¹⁶."

-Enhanced membrane binding of Δ ABD:

i) Membrane binding by FRET: did the author perform a control test with p85 alone in the presence of the membranes? I am curious about the contribution of the presence of p85 to the FRET signal quantified in these experiments.

This is an excellent point raised by the reviewer. There have been previous experiments from the Cho lab that has indicated the role of the nSH2 and cSH2 of p85 in membrane binding.

To address this point and unambiguously determine the role of p85 in membrane binding, we have now expressed and purified full-length p85 α alone in addition to the complex with p110 α . We performed Protein-Lipid FRET that measured membrane recruitment of p85 α under basal and pY bound conditions.

The presence of p85 α in FRET experiments at the same concentration as in the p110 α - p85 α led to a FRET signal significantly lower than for the p110/p85 complex, with no increase at all upon pY binding (see new Fig 2). The values obtained for this experiment were in a similar range to those with the delta C-terminus of p110 (Δ C).

This suggests that there is a limited effect of p85 in the membrane binding FRET signal. We have added additional text regarding this experiment to the results section (see paragraph below):

“To determine the role of free p85 α in PI3K membrane recruitment, we also purified recombinant free p85 α and analyzed the protein-lipid FRET signal. There was a weak FRET signal for p85 α alone, with no change upon pY binding. This signal was significantly lower than the p110 α /p85 α complex (Fig S4F), indicating a limited role of p85 α in the FRET signal, and suggests that membrane binding of PI3K is mainly driven by interactions with the p110 α catalytic core.”

ii) Is the enhanced membrane binding only linked to the removal of the ABD domain or to both ABD domain and p85? In other words, was the implication of p85 to the WT p110 α membrane binding assessed?

This is an excellent question from the reviewer, and one we very much would like to know the answer to. Unfortunately due to biochemical limitations this is an extremely challenging question to answer.

The issue that arises is that we are not able to purify the full length p110 α protein alone, as the ABD domain seems to promote aggregation (likely due to the hydrophobic iSH2 interacting surface being exposed). Therefore we can only generate a construct lacking both the ABD and the p85, making it impossible to specifically answer the question on which plays a more important role in blocking membrane binding of p110.

Clearly our new data shows that p85 alone does not appreciably bind membranes. This combined with data showing that the free catalytic core lacking the ABD and p85 binds membranes significantly better than pY activated wild-type. However, this does not unambiguously define if the ABD blocks binding or the iSH2 blocks binding (nSH2 will be removed by pY, which is why there is enhanced binding with pY for the complex). Although both the ABD and the iSH2 of p85 interact with regions of p110 that bind membranes in the free core (see more below).

-Membrane binding surface of Δ ABD:

while the results at the N- and C-lobe are clearly convincing, I am not quite sure about the slight HDX difference in one of the identified peptides in the C2 region. Further experiments (for instance, effect of a critical point mutation in this region on binding to p85) could consolidate the drawn conclusion related to the impact of the C2 domain and therefore the disengagement of p85 in membrane binding.

This is a good point raised by the reviewer and requires some additional information on the fundamental basis of H/D exchange. The differences seen in the N-lobe and C-lobe are in regions that either have intrinsic secondary structure, or regions that putatively undergo disorder-order transitions upon membrane binding (C-terminus). The region in the C2 domain that is protected upon membrane binding is in a loop that likely has no intrinsic secondary structure. From examining the peptide incorporation data for this region (343-354, Fig S4D) there is only protection at early time points.

Attempts to generate C2 mutant versions of the catalytic core of p110 α were unsuccessful, preventing our ability to test this further.

To more clearly illustrate the H/DX point we added the following addition to the results

'The largest differences occurred in the C-terminus, and N-lobe, with only minor differences in the C2 domain. However, the region of the C2 domain that interacts with membrane has limited secondary structure (see Fig S4G), which can make tracking transient membrane differences using HDX challenging. Previous HDX-MS experiments testing N345K p110 α -p85 α binding to membranes showed this same region being protected by membranes¹⁶.'

And figure legend of Fig. S4D

'Note that the intrinsic exchange rate of different regions explains some of the differences in H/D exchange seen upon membrane binding. Regions with stable secondary structure in the absence of membrane are protected primarily at later time point (this is due to membrane binding further stabilising the secondary structure, see 735-744). The C-terminus undergoes a putative disorder-order transition (1056-1068), and shows stabilisation at all time points, with rapid exchange in the absence of membranes. Finally, regions with limited secondary structure (343-354) show protection at only early timepoints of D₂O exchange.'

-Role of the conformation of the C-terminus in PI3K activation:

This part does not provide new experimental data but rather a new analysis of already published data.

We agree that this section was overly long in discussing previously published crystal structures and have truncated this section extensively, removing roughly half the words.

-Conformational changes in oncogenic mutations C-terminus:

Impact of the Δ Cter and H1047 mutations in the 1014-1021 peptide are very subtle compared to the observed differences in the rest of the p110 α sequence. In addition, the differences are observed mainly for the longest incubation timepoint, as opposed to the other peptides shown in figure S7. Could the authors comment on that? How comparable are the differences observed in this region and the other ones showing a more obvious deuteration difference (e.g. in the C2 region)?

This is an excellent point by the reviewer, and once again requires explanation of some of the insight into the intrinsic dynamic of different regions that HDX analysis can provide. It is not unusual to see regions where only a single timepoint at long exposures show differences. This is indicative of a region with amides that are highly protected in the Apo

state (rigid secondary structure), and in the pY bound state are slightly destabilised (i.e. changes that only occur at the latest time points).

The C2 domain regions are not as intrinsically stable and hence show differences throughout the entire time course of deuterium exchange (see peptide 444-475).

In both cases the deuterium differences are comparable, they just indicate a difference in the dynamics of the region that shows changes in exchange.

We have added additional details regarding this to the legend of Fig S7:

Note that the intrinsic exchange rate of different regions explains some of the differences in H/D exchange seen upon pY binding. Regions with stable secondary structure in the absence of pY are protected primarily at later time points (this is due to pY binding further stabilising the secondary structure, see 1014-1021). Regions with less stable secondary structure show changes throughout the time course (see peptide 444-475).

-In figure S3:

i) Sequence coverage of p85A: Why are there many missing peptides (mainly in the SH2 in p85) in the presence of the membrane?

We apologise for the oversight, one of the graphs here had a x axis bar with a dotted line that looked like HDX data. This has been removed. All experiments in Fig S3 have the same number of peptides. The experiment comparing the kinase dead and kinase active PI3K were done with a slightly lower concentration of protein (as this was a control experiment, and we mainly wanted to check that membrane binding regions behaved the same). For this reason, there was decreased coverage in the iSH2 domain (which is the region of the protein that has multiple peptides with intensity close to the signal/noise threshold).

ii) Why only 2 deuteration timepoints are shown for the different peptides in panel F?

Once again as this is a control to validate that kinase dead (KD) and kinase active (KA) behave similarly upon membrane binding we chose two timepoints that we know from previous experiments cover all significant changes upon membrane binding for WT PI3K. Hopefully it is clear from the overlap of the triplicate HDX curves that this change is the same, which is further validated by the almost exactly the same protein-lipid FRET signal between KA and KD(Fig. S3B).

-Table S1: how come number of peptides/length and redundancy are the same between WT and Δ ABD PI3K?

We apologise for the confusion on this point. this data is only looking at peptides that are conserved between the catalytic core and the full length p110 alpha. While there are additional peptides that could be mapped in the full length, there would be no possible comparison in the core. For this reason, we have the same size of analyzed peptides in both data sets (see source data file for the full list of peptides). To clarify this, We have included more details on this in the methods, and in the Table S1 legend.

-Please add the state info (State A – State B =?) in all figures showing the #D Difference. We have added these changes to all #D difference figures

-Data present in figure S4 related to main figure 2B is previously published? If so, please specify in figure legend.

None of the data in Fig S4 is previously published, with no panels referring to figure 2B.

-Overall, some of the main figures are densely packed (like for instance panels C to E in figure 3); consider revising them to make them clearer.

We have resized Figure 3+5 to make them more clear to readers

-Below some suggestions and questions for Figure 1:

i) Increase the size of the scheme in panel A.

We agree that this makes the figure more legible, and have increased the size of this schematic in panel A.

ii) Is the regulatory motif the part coloured in green or inside the dashed box? Better stick to colouring the regulatory motif and removing the dashed box.

We agree and have removed the dashed box.

iii) What do the green circles refer to in Panel B structure? consider changing the green colour since it's the colour code chosen for the regulatory motif (unless it's related to that motif).

There are currently no green circles in our figures. This could be a misunderstanding because of yellow colored mutation residues in C2 and helical domain that are buried behind the transparent blue p85. These regions underneath may look like a different color. We have added more description about p85 being shown as a transparent surface that explains this potential color issue.

iv) What does the red arrow relate to in panel E?

Additional information related to the questions above could be added in the figure legend for more clarity.

We agree and have clarified that the arrow indicates the reorientation of the C-terminus that occurs in the H1047R mutant.

v) Use the same magenta colour used in the mapped structures for the C-terminus in the schematic in panel A.

We thank the reviewer for the suggestion. We have changed the color to be Magenta and consistent in all panels.

Reviewers' Comments:

Reviewer #2:

Remarks to the Author:

The authors have addressed all of my concerns in the revised version.